# Post-training for Efficient Communication via Convention Formation

**Yilun Hua, Evan Wang, and Yoav Artzi**
Department of Computer Science and Cornell Tech
Cornell University
{yilunhua, yoav}@cs.cornell.edu, ew447@cornell.edu

## Abstract

Humans communicate with increasing efficiency in multi-turn interactions, by adapting their language and forming ad-hoc conventions. In contrast, prior work shows that LLMs do not naturally show this behavior. We develop a post-training process to develop this ability through targeted fine-tuning on heuristically identified demonstrations of convention formation. We evaluate with two new benchmarks focused on this capability. First, we design a focused, cognitively-motivated interaction benchmark that consistently elicits strong convention formation trends in humans. Second, we create a new document-grounded reference completion task that reflects in-the-wild convention formation behavior. Our studies show significantly improved convention formation abilities in post-trained LLMs across the two evaluation methods.

## 1 Introduction

Humans naturally display rapid adaptation during linguistic interactions by developing increasingly efficient ways to refer to concepts. This formation of ad-hoc linguistic conventions has been repeatedly observed in studies (Krauss & Weinheimer, 1964; Brennan & Clark, 1996; Hawkins et al., 2020a), and is a cornerstone of naturalistic human language interaction. It not only improves the accuracy of relaying information, but also reduces its costs. Contemporary large language models (LLMs), on the other hand, do not naturally show this behavior (Hua & Artzi, 2024).

We propose a targeted post-training process to develop this general ability in LLMs, such that models spontaneously form conventions as an in-context behavior. We heuristically extract examples of convention formation from human corpora to construct minimal pairs, where the minimal difference between two paired examples is in the demonstration of this behavior. In addition, we enhance the reasoning process of model by introducing reference planning tokens, which mark when a referent is a re-mention. We treat this data as preference pairs for DPO-style policy optimization (Rafailov et al., 2024), and carefully design the optimization process to acquire generalizable convention formation ability.

We design our evaluation process to differ from the training in both data and scenario, with the aim of quantifying the generalizable change in model behavior. We create two evaluation tasks. In the first, we follow the reference game design from cognitive science studies (Krauss & Weinheimer, 1964; 1966; Clark & Wilkes-Gibbs, 1986; Hawkins et al., 2020a), where a speaker repeatedly directs a listener to select a target from a shared set of items. Reference games were designed to show strong effects in humans with relatively short interaction and limited data. While reference games often rely on visual stimuli, we design a new text-only variant to avoid any confounding issues due to the visual reasoning abilities of models. This broadly applicable new variant shows similar trends to visual reference games in baseline human studies we perform. The second evaluation scenario we design is a document-grounded utterance completion task, rooted in scenarios where an assistant uses a reference document to answer a user's questions and explain concepts. This task complements the reference games with a format inspired by retrieval augmented generation, where we assume the model is generating a response based on given references.

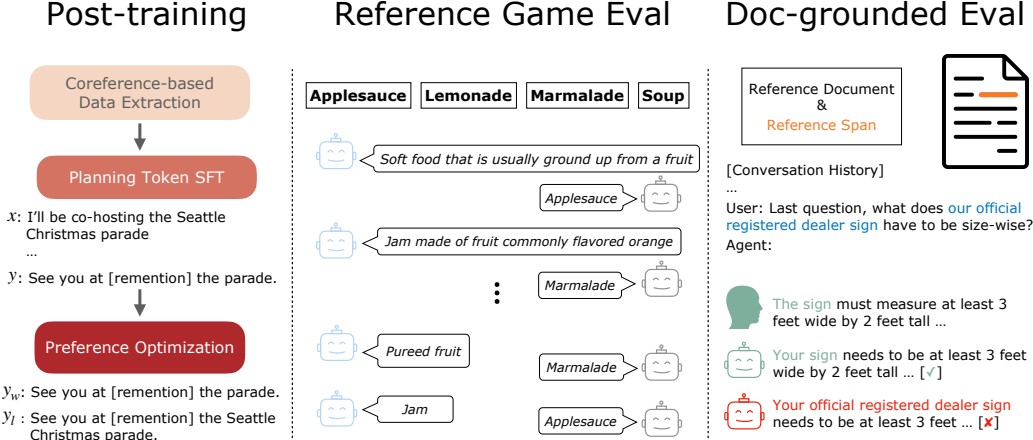

Figure 1: Overview of our framework. We propose a post-training method to improve LLMs' convention formation (left) and two evaluation tasks (middle and right) covering different interaction scenarios and domains.

We study several state-of-the-art proprietary and open-source LLMs and find that, even in the text-only domain, LLMs still lack the ability to develop ad-hoc concise referring expressions and form conventions. We apply our post-training method to the open-source models and observe significant improvements. The post-trained LLMs shorten their messages by up to 26% throughout the reference games, maintain message consistency to form conventions, and allow the listener to identify the target with increasing accuracy. In the document-grounded task, the post-trained models also outperform their off-the-shelf counterparts by a substantial margin. All these improvements appear even though the model is only trained on data from natural conversations and never on the evaluation scenarios.

Overall, our contributions are two-fold. First, our tasks form new avenues to assess the communicative efficiency and adaptability of LLMs, without the potential confounding factor of visual stimuli. Second, we show that contemporary LLMs lack this ability for efficient communication, and design a post-training method to mitigate this issue. Our methods lay a foundation for research to further improve LLMs' convention formation and communication efficiency. Our code is available at https://github.com/lil-lab/post-train-for-efficient-communication.git.

## 2 Background and Related Work

**Repeated Reference Games** Repeated reference games with visual stimuli are widely used to study human convention formation (Krauss & Weinheimer, 1964; 1966; Clark & Wilkes-Gibbs, 1986; Hawkins et al., 2020a). A standard repeated reference game involves a speaker and a listener interacting over many trials and a recurring set of referents (images). In each trial, a target referent is revealed only to the speaker, who needs to produce a description to guide the listener to select the target. The two participants see the referents in different orders and cannot use the position information in their referring expression (e.g., cannot say *pick the first item*). After each trial concludes, the speaker sees the item selected by the listener and the listener knows if their selection is correct. As the participants complete more trials together, the same referent will repeatedly appear as the target. This has been observed to lead the speaker to use increasingly efficient language, the listener to become increasingly accurate in identifying the target. Hua & Artzi (2024) adapts the repeated image reference games for multimodal LLM evaluation. They show that multimodal LLMs struggle with developing more concise referring expressions as the speaker, but achieve near-human performance as listeners. Therefore, they use GPT4 as the listener to automate the evaluation of LLM speakers, a design choice we also adopt.

**Convention Formation** Humans are inclined to minimize the effort used in communication, both the effort one spends to articulate a piece of information and the effort their partner needs to understand it (e.g., Zipf, 1949; Gibson et al., 2019; Yin et al., 2024). This principle is widely observed in conversations, in the form of convention formation (Krauss & Weinheimer, 1964; 1966; Clark & Wilkes-Gibbs, 1986; Hawkins et al., 2020a; Haber et al., 2019), when the referring expression for the same item shortens and the two interlocutors arrive at a mutually understood, concise expression that they will stick to for the rest of the interaction. Importantly, the expression for an item quickly becomes consistent – a speaker usually keeps dropping words from their message but rarely introduces new words to refer to the same item (Hawkins et al., 2020a). Hua & Artzi (2024) observed that LLMs do not exhibit this trend. Bringing about this property of convention formation is a critical part of our objective.

**Learning for Language Adaptation in Agents** Prior works on systems that adapt their language for efficient communication have primarily focused on training models for a fixed, specific interaction task and using data from the task. Hawkins et al. (2020b) applies a continual learning method to elicit language shortening within repeated reference games. Takmaz et al. (2020) trains a model on data from an image reference task, PhotoBook (Haber et al., 2019), and shows that models trained to ground on the image and conversation history can generate more efficient referring expressions for this task. Similar paradigms have also been adopted to study models' adaptation to interlocutors with different background information in reference games (Greco et al., 2022). In contrast, our goal is to train a general-purpose LLM that can show general language efficiency adaptation, whether in a highly controlled lab-like game scenario or scenarios close to how these models are used in production. Importantly, using only the data extracted from natural human conversations, our post-training method significantly improves LLMs' convention formation ability in the repeated reference game, even though we do not train them on this specific task format.

## 3 Post-training for Ad-hoc Convention Formation

We propose a post-training method that includes several stages: (a) constructing preference data from heuristically identified examples of convention formation in human conversations; (b) adding a special planning token to enhance explicit reasoning about references; and (c) policy optimization using the preference pairs. Throughout, we design learning to preserve the model's general capabilities. Appendix B includes additional implementation details.

**Preference Data Construction** We use a coreference resolution (Bagga, 1998) model on scripts of TV series, a domain of text that is rich with conversational interaction, to identify repeating instances of convention formation, and modify the data to create preference pairs. Coreference resolution generates reference chains, where mentions that refer to the same entity appear in their order in the text. Such chains often display convention formation, with later references being shorter and more consistent.

We heuristically identify examples where a concept is initially referred to (i.e., mentioned) with a noun phrase in an utterance $i$ and is re-mentioned in a later utterance $j$ with a more concise referring expression. Each such example may also include intermediate re-mentions, meaning that the re-mention in utterance $j$ is not the first re-mention. Therefore, each reference chain can provide multiple demonstrations. This is important because it shows to the model that the desired behavior is not only in the first re-mention, but persists over the entire reference chain.

We use this data to construct triples of $(x, y_w, y_l)$, where $x$ is the conversational context (i.e., the history of the interaction), $y_w$ is the preferred continuation, and $y_l$ is the dis-preferred continuation. We set $x$ to be the conversation history until a reference, and $y_w$ and $y_l$ to represent desired and undesired utterance to contain the reference. We create two types of preference pairs. The first type is a demonstration of the observed convention behavior. We set $y_w$ to be the observed re-mention text, which reflects convention formation, and $y_l$ to use the more verbose first-mention text as the re-mention. This aims to suppress the behavior of simply repeating mentions verbatim, and encourage adjusting them as expected when

conventions can be formed. The second type aims to preserve how the first mentions are generated, because the model must avoid a faux conventionalization behavior in the first mention, where it has no shared common ground. We set $y_w$ to the original first mention, and $y_l$ to the conventionalized re-mention that was observed in utterance $j$. We extract the first type of pairs from reference chains of at least two re-mentions, to better demonstrate the convention formation process. For the second type, we extract from reference chains of at least one re-mention. We process 2,000 TV scripts from Chen et al. (2022), and the total numbers of extracted first and second types of pairs are 11,106 and 10,135 respectively.

**Adding Mention Planning Tokens**   We further modify the preference pairs to explicitly reflect the distinction between mentions and re-mentions using a special token: re-mentions are preceded by a [remention] token. We expect this explicit marking to allow the model to better separate its treatment of initial and later mentions. This separation could allow the model to develop convention formation skills, without hurting how initial mentions are generated. Beyond adding the new token to existing preference pairs where appropriate, we also create new preference pairs focused on training the model to use this token. We add triplets $(x, y_w, y_l)$, where if $y_w$ captures a re-mention, then $y_w$ contains the corresponding planning token and $y_l$ does not. Conversely, if $y_w$ captures a first-mention, then $y_w$ does not contain the planning token and $y_l$ contains a misplaced planning token preceding the first-mention. Empirically, we observe the planning token to help (Section 6). Figure 9 in Appendix E.1 shows an example training instance.

**Regularized SFT for Planning Token Learning**   We precede preference optimization with a supervised finetuning (SFT) phase to help the model use the [remention] token properly. Given a training preference pair for a re-mention $(x, y_w, y_l)$, we construct a new SFT input-output pair $(x, y_w)$, where $y_w$ is the target utterance with the special planning token [remention] inserted before the re-mention. To account for the new [remention], we extend the input and output embedding tensors of the LLM as needed. We design the training process carefully to learn the planning tokens while avoiding overfitting. Instead of the standard cross-entropy language modeling loss, we calculate the cross-entropy loss on the planning tokens only and calculate a regularization loss on the rest of the tokens. Let $N$ be the set of all tokens in $y$, $M$ be the set of planning tokens in $y$, our loss for a training instance $(x, y)$ is[1]

$$\mathcal{L}_{SFT} = -\frac{1}{|M|} \sum_{t \in M} \log \pi_\theta(y_t | y_{<t}, x) + \frac{1}{|N| - |M|} \omega \sum_{t \notin M} \mathcal{D}_{JSD}(\pi_\theta(y_t | y_{<t}, x) || \pi_{\text{ref}}(y_t | y_{<t}, x)) \quad (1)$$

$$\text{s.t. } \mathcal{D}_{JSD}(\pi_\theta || \pi_{\text{ref}}) = \frac{1}{2}\mathcal{D}_{KL}(\pi_\theta || \frac{\pi_\theta - \pi_{\text{ref}}}{2}) + \frac{1}{2}\mathcal{D}_{KL}(\pi_{\text{ref}} || \frac{\pi_\theta - \pi_{\text{ref}}}{2}) \quad ,$$

where $\omega$ is the regularization coefficient. $\pi_{\text{ref}}$ is the original weights of the model. $\mathcal{D}_{JSD}$ is the Jensen-Shannon divergence (JSD), which is a symmetric and bounded alternative to KL-divergence. We empirically observed that JSD provides more stable learning.

**Preference Optimization**   We use all preference data $(x, y_w, y_l)$ for this training stage. we use the APO-zero loss (D'Oosterlinck et al., 2024), which directly encourages an increased winning example likelihood and a decreased losing example likelihood. We opt for APO-zero over the standard DPO loss (Rafailov et al., 2024), because DPO has been shown to sometimes cause the winning example likelihood to decrease during training and thereby hurts the performance, an issue known as likelihood displacement (Pal et al., 2024; Razin et al., 2025). The minimal difference between our preference pairs (i.e,. by a single noun phrase or re-mention token) is a known cause to likelihood displacement. Our loss for this training stage is:

$$\mathcal{L}_{APO} = -\sigma\left(r_\theta(x, y_w)\right) + \sigma\left(r_\theta(x, y_l)\right) \quad , \quad (2)$$

where $r_\theta(x, y_w) = \beta \log \frac{\pi_\theta(y|x)}{\pi_{\text{ref}}(y|x)}$ and $\beta$ is a hyperparameter similar to DPO's $\beta$.

---

[1]We use a somewhat non-conventional set notation here because it simplifies the overall notation, and does not require testing for [remention] tokens in the objective.

## 4 Evaluation Tasks

We design our evaluation tasks to be separate and distinct from our training regime, with the aim of assessing general abilities beyond the training data. Figure 1 illustrates the two evaluation tasks. The two tasks approach the evaluation in different ways, and are complementary to each other.

### 4.1 Text-only Reference Game

Reference games with visual stimuli are commonly used to study convention formation in human interaction (Krauss & Weinheimer, 1964; 1966; Clark & Wilkes-Gibbs, 1986; Hawkins et al., 2020a) and more recently in multimodal LLMs (Hua & Artzi, 2024). We create a new text-only variant, loosely inspired by the Taboo game.[2] The dynamics are identical to conventional reference games. There are two participants: a speaker and a listener. They both observe a set of potential referents (i.e., as context). At each turn, the speaker is asked to describe a target for the listener to select. The turn ends successfully if the listener selects the right target. Otherwise, it ends with a failure. Instead of images, we design the set of potential referents to be textual names of everyday items. To avoid the degenerate behavior of simply copying the word or part of it (e.g., using *pan* to refer to *dustpan*), we restrict the use of the word and its lexemes. For example, a valid message for the target *dustpan* can be *the flat, rectangular thing with the handle that you sweep into*, but not *the pan*.

Similar to image reference games, we use a repeated version of the game to study convention formation, where each interaction includes multiple turns with repeated targets. Figure 10 and Figure 16 in Appendix E.1 show an example model query and messages evolving as the interaction progresses in our reference game. This evaluation task uses a relatively synthetic interaction scenario, but aims to emphasize the behavior we aim to study, so measurements of trends are clear and significant.

We follow prior work to evaluate convention formation, including tracking utterance length (e.g., Hawkins et al., 2020a;b) and tracking message consistency with word novelty distance (WND; Hua & Artzi, 2024), which tests the lexical stability of consecutive references to the same item (i.e., if new words are introduced). In our experiments, we conduct human studies and confirm that the scenario elicits the expected convention formation trends in human interaction (Section 6, gray curves in Figure 2). Indeed, human text-only games show trends similar to those observed in image-based games: turn success increases, utterance length decreases, and WND decreases.

We extract candidate referents from a dataset of Taboo games.[3] We create challenging reference sets by sampling related items by clustering similar words using their embeddings.[4] Each set includes four referents. A similar method for constructing challenging reference sets was used for image-based reference games (Hawkins et al., 2020b). We extract 47 referential contexts for evaluation and adopt the same game length as prior studies: 24 turns (trials) organized into six repetitions. Each repetition includes a turn for each item in the set.

### 4.2 Document-grounded Utterance Completion

The second task is based on the Doc2Dial dataset (Feng et al., 2020), where a user is querying an agent in a document-grounded interaction. At each step, the user asks a question, and the agent responds based on the document. Doc2Dial interactions also include the span of text in the document used to produce each agent utterance. As with other human interactions, Doc2Dial dialogues show convention formation when referents are repeated.

Our task requires the model to complete the response of a single turn in a Doc2Dial interaction. The model is given both a snapshot of a human conversation transcript at a specific turn following a user question and the text span from the document that contains

---

[2]https://en.wikipedia.org/wiki/Taboo_(game)
[3]https://github.com/Kovah/Taboo-Data
[4]We use the OpenAI text-embedding-3-small embedding model.

the information to the answer. The model is expected to generate the answer, based on the document, the reference span, and the history of the interaction. Figure 11 in Appendix E.1 provides an example input. If the target re-mention phrase is not in the beginning of the original human agent response, we pre-fill the model's generation with the first part of the human response such that the model would generate from where the re-mention phrase would start. Appendix B provides more implementation detail.

We create the test instance by processing the Doc2Dial dialogues similar to how we prepare the training data in Section 3. We extract examples where the assistant refers to a previously mentioned concept with a more concise expression, at least two words shorter. This process generates a total of 481 evaluation instances. This scenario is closer to our training setup, but the interaction data differs significantly. While we use TV scripts to create our training data, Doc2Dial includes information-seeking interactions in four specific domains (social security, veterans affairs, motor vehicles, and public health).

We evaluate model performance as win-rate against another model or human ground truth. The win-rate is computed using a GPT-4o judge that is instructed step-by-step to parse the two utterances and judge if the utterance produced by the evaluated LLM uses more, less, or equally concise referring expressions compared to the counterpart model or human utterance. Figure 12 in Appendix E.1 shows the GPT-4o judge prompt. Appendix B presents more implementation details and the GPT judge's agreement with human annotation.

The document-grounded utterance completion evaluation differs significantly from the text-based reference game task. It does not include repeated interactions, and therefore does not evaluate interaction trends. Instead, it examines the behavior in more realistic scenarios, where the context (i.e., the document and the interaction history) is more complex.

# 5 Experimental Setup

**Text-only Reference Games**   We automate our evaluation by using GPT4o-mini as the listener, which is efficient to run, has a low cost, and shows similar performance to human listeners based on our pilot study (Appendix A) and prior work (Hua & Artzi, 2024).

As part of our experiments, we collect human interaction data to quantify how our new setup elicits convention formation in human speakers. We use the crowd-sourcing platform Prolific, and collect 200 interactions with human speakers and GPT4o listeners. We design the instruction and format of the study such that the human participants do not know they are interacting with a model, thus eliciting more natural human behaviors. The only goal for the participants is to help the listener identify the target correctly, and a participant receives a monetary bonus every time their partner successfully identifies the target. We pay 2.6 USD per interaction, with a bonus of 0.03 USD for each successful turn.

To minimize potential priming effects, the instructions for the participants do not mention "adaptation" or "communication efficiency", and the participants can only complete the interaction once. A side-effect of this light instruction is that some participants misinterpreted the instruction, and concluded that they are required to change their referring expressions between turns. Naturally, this led to an increase in WND and sabotaged the creation of conventions. We identify this group by fitting Gaussian mixture models (GMM) on the WND scores. We remove the low-consistency component (35.5% of the participants), and only use the remaining users for evaluation. This latter group displays clear trends of convention formation, the behavior we aim for our models to emulate. We further discuss our human studies in Appendix A.

**Models**   We evaluate Claude-3.5-sonnet, GPT4o, Gemma-2-9b-instruct, and Llama-3.1-8b-instruct. We apply our post-training to the open-weight models, Gemma and Llama.

**Training**   We use the coreference resolution model from Zhang et al. (2023) to prepare the data. We use LoRA for both the SFT and APO stages. In the SFT stage, we additionally update the embeddings of the added token while keeping the original vocabulary's embeddings frozen. Appendix B provides hyperparameters and implementation details.

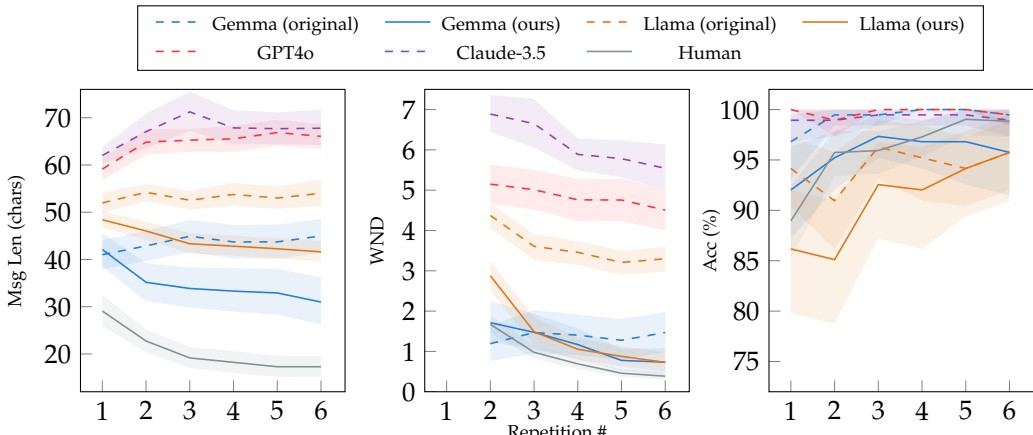

Figure 2: Reference game results. Shaded areas are bootstrapped 95% confidence intervals.

**Ablations** We experiment with several ablations, using the same training schedule and hyperparameter search as our post-training method whenever possible: (a) **APO only**: APO-zero training on the preference data directly, no planning tokens or SFT updates are used; (b) **SFT**: SFT on the inputs and target completions from the SFT stage (Section 3), but with planning tokens removed and using a standard language modeling loss; (c) **Ours w/o planning tokens**: standard SFT and language modeling loss to initialize the model with a small number of updates and then adopt APO training; and (d) **Ours w/o JSD**: standard SFT on the input and target utterances that include the planning tokens, without adding the JSD loss, and with standard language modeling loss calculated on all the tokens of the target output, followed by the same APO training stage as our main method.

## 6 Results

We observe significant deficiencies in convention formation in all the off-the-shelf models we evaluate across both evaluation testbeds. Our post-training treatment significantly improves convention formation behavior. Figure 2 shows convention formation trends on the reference game scenario, Table 1 provides the grounded-document scenario results, and Appendix C provides ablation analysis.

### 6.1 Text-only Reference Games

Figure 2 shows the results for the reference game evaluation, where we report the average message length in characters, Word Novelty Distance (WND), and average accuracy across the six repetitions of the interaction. All off-the-shelf LLMs fail to form conventions or improve their communication efficiency. They often increase message lengths over time and introduce new words to refer to repeated items, degrading consistency (higher WND).

Our post-trained models show statistically significant improvement in language efficiency. Gemma (ours) shortens its messages by 26% (42.1 characters → 31.0 characters) from Repetition 1 to Repetition 6 and Llama (ours) by 14% (48.4 characters → to 41.6 characters). The post-trained models are also substantially more consistent. Their WNDs consistently decrease as the game progresses, reaching below 1 at the end of the interaction. This means that the post-trained models' messages gradually stabilize and conventionalize.

We observe that a small accuracy gap exists between post-trained and original models in Repetition 1. Interestingly, this gap makes the post-trained models' accuracies closer to that of humans at Repetition 1. More importantly, the post-trained models' accuracies improve over time while their messages shorten, eventually using substantially more concise language to achieve accuracies close to the original models.

The human convention formation trends we observe validate our new text-based reference game setup, and closely follow trends observed in past work using visual stimuli (Hawkins

| Post-trained vs. Original | # Wins | | | | Competence |
|---|---|---|---|---|---|
| | Ours | Original | Tie | Cannot Decide | Compet. Rate (%) |
| Gemma | **121** | 45 | 299 | 16 | - |
| Llama | **135** | 46 | 291 | 9 | - |
| **LLM vs. Human** | **# Wins** | | | | **Competence** |
| | LLM | Human | Tie | Cannot Decide | Compet. Rate (%) |
| Gemma (ours) | **109** | 97 | 259 | 16 | **79.1** |
| Gemma (original) | 88 | **130** | 244 | 19 | 71.9 |
| Llama (ours) | 90 | **114** | 257 | 20 | 75.3 |
| Llama (original) | 83 | **160** | 227 | 11 | 66.0 |
| Claude | 76 | **168** | 220 | 17 | 63.8 |
| GPT | 73 | **192** | 203 | 13 | 59.0 |

Table 1: Document-grounded utterance completion main results. We report win-rate compared to humans and the original off-the-shelf model. In rare cases, one model mentions a different concept than the other does or fails to re-mention any concepts. In this case, the GPT4o judge would output "cannot decide."

et al., 2020a;b). This demonstrates that our formulation is a fitting interaction scenario to study convention formation, not only in models, but also in humans.

Overall, the results show a significant increase in the alignment between model and human behavior. Humans still show better communication efficiency improvement than any LLM, eventually reaching 17.3 characters and 0.38 WND. The remaining gap indicates the problem is not completely solved, despite the significant improvement by our method. Appendix E.2 presents examples of successful convention formation as well as negative examples.

## 6.2 Document-grounded Utterance Completion

Table 1 reports the utterance completion results. In addition to the GPT4o-judged win rate, we also report the competence rate, a single number for quick comparison:

$$\text{competence rate (\%)} = \frac{\text{\# model wins} + \text{\# ties}}{\text{\# model wins} + \text{\# ties} + \text{\#human wins}} \quad (3)$$

To compare a specific post-trained model and its off-the-shelf counterpart, we conduct pairwise evaluation on them and directly look at how often one model wins or ties with the other model. This allows for a more finegrained comparison because a post-trained model may show greater shortening than the original model, but the human utterance shows even more shortening. These instances will not be reflected by the models' competence rates, which are based on win and tie rates with humans.

Compared with the original models, our post-trained models more often use a more concise referring expression as the re-mention than the other way around. Llama shows the greatest improvement; the post-trained model wins in 135 instances and the original model only wins in 46 instances. The gaps between the post-trained and original models are statistically significant with $p < 0.01$ under a sign test. Our post-trained models' wins are also substantially closer to humans' than the original ones, and achieve higher competence rates.

The weak performance of off-the-shelf models suggests that contemporary LLMs are suboptimal in adapting their language for efficient communication. This result further supports our conclusions from the lab-like setting of reference games and manifests these models' issues in a scenario closer to how LLMs are used in production.

## 6.3 Additional Experiments and Analyses

**Ablations and Prompting-based Attempts** We also experiment with ablating the [remention] tokens, Jensen-Shannon Divergence (JSD) Loss, and the SFT stage. The experiments show that all these components are necessary for the model to learn generalizable convention formation abilities and achieve balanced improvements across tasks. Appendix C discusses our ablation studies in detail.

| Model | Δ MixEval (%) |
|---|---|
| Gemma (ours) | +0.2 |
| Llama (ours) | -0.8 |
| Nvidia Nemotron | -6.3 |
| Mazzaccara et al. (2024) | -1.5 |

Table 2: Impact of post-training on general LLM capabilities, as measured in MixEval-hard score change.

| % of Occurrence | Drop Phrase | Rephrase | New Feature |
|---|---|---|---|
| **Human** | 66 | 19 | 15 |
| **Gemma (ours)** | 62 | 11 | 26 |
| **Llama (ours)** | 43 | 35 | 21 |

Table 3: Language strategies of efficiency adaptation in the reference games.

Additionally, we experiment with eliciting convention formation with prompting. Hua & Artzi (2024) has shown that prompting is not a satisfactory solution to the convention formation problem, with their results from image-reference games. We conduct similar prompting experiments with our text-only tasks. Overall, our conclusions are similar. Prompting and few-shot examples do not work well in eliciting convention formation. Appendix D presents the results of these experiments.

**Preserving the LLM Capabilities** We evaluate our models on MixEval-hard (Ni et al., 2024), a benchmark for general LLM capabilities, which consists of diverse tasks and has state-of-the-art correlation with human judgment. We calculate the change in MixEval score (an accuracy-based score out of 100) after post-training. We find that our approach led to no degradation on Gemma, and only a small decrease of 0.8 on Llama. To better interpret the magnitude of this decrease, we additionally evaluate two other post-trained models and compute their MixEval score changes. Importantly, because we just focus on enhancing model's convention formation ability, our method is not comparable with common post-training works that intend to enhance model's general abilities and cover diverse datasets/tasks. Instead, we compare with post-training interventions that are intended to add or enhance specific skills. We compare against Nvidia Nemotron Nano,[5] a post-trained version of Llama-3.1-8b-instruct, where the post-training aims to enhance the math and reasoning capabilities. We also compare with the model from Mazzaccara et al. (2024), who post-train a Llama-2-7b-chat to enhance its ability to ask informative questions. Compared to post-training interventions for these models, our method does not cause a substantial performance decrease (Table 2).

**Analysis of Language Characteristics of Adaptation** We analyze humans' and post-trained LLMs' language changes from one repetition to another in 30 reference games, and identify three categories of adaptation behaviors. Table 3 presents the results. The most common strategy for both humans and the post-trained LLMs is to remove a phrase from the previous message while keeping the rest of the message the same ("Drop Phrase"). The removed phrase is oftentimes a prepositional phrase or a clause. Another strategy is rephrasing, using a concise expression to replace part or all of a message while conveying a similar meaning. For example, the referring expression for a *waterbed* can change from *the mattress that is filled with liquid* to *the liquid-filled mattress*. Finally, both humans and LLMs sometimes replace the previous description by describing a new feature that is more concise. For example, a hammer can be initially described as *the tool that is used to fix things around the house* and later as *the tool that is used to build things*. These analyses show that humans and LLMs share similarities in the strategies they use, though LLMs' more frequent use of new features and/or rephrasing likely contributes to their slightly lower language consistency compared with humans (as their WNDs show).

---

[5] https://huggingface.co/nvidia/Llama-3.1-Nemotron-Nano-8B-v1

| % of Words Reduction | NOUN | VERB | ADJ | ADP | PRON |
|---|---|---|---|---|---|
| **Human** | 15 | 13 | 7 | 14 | 18 |
| **Gemma (ours)** | 18 | 14 | 33 | 8 | 0 |
| **Llama (ours)** | 34 | 8 | 24 | 16 | -3 |

Table 4: Reduction of POS counts from Repetition 1 to Repetition 6 in the reference games (percentage of the total word reduction). Negative value means the count increased.

Additionally, we examine how the numbers of common POS tags change for models and humans over repetitions in the reference games (Table 4). For all systems, we see substantial reduction in the number of nouns, verbs, and adpositions. Notably, the reduction in adjective counts accounts for a larger proportion of the overall length reduction in Gemma. This is because the original Gemma model would start the game with longer messages that contain more adjectives than humans. Because our post-training is not intended to alter the model behavior for initial mentions, the post-trained model's messages still often contain adjectives in Repetition 1 (average adjectives count: 1.5). Humans, however, start with relatively concise language with only an average of 0.5 adjectives in Repetition 1. Another difference is that human messages more often contain pronouns in early repetitions, which are then dropped later (pronoun count in Repetition 1: 0.7 (human), 0.5 (Gemma)). For Llama, the shortenings in nouns accounted for a greater proportion (34%) of reduction, also because the post-trained Llama (just like the original Llama) started the interaction with more nouns than humans did (average noun counts: 2.7 (Llama), 1.5 (Human)).

## 7 Conclusion, Future Work, and Limitations

We introduce a post-training method for LLMs to acquire convention formation abilities and thereby improve their naturalness and communication efficiency in multi-turn interactions. We also design two text-only tasks to evaluate convention formation, without the potential confounding factor of visual stimuli present in prior work. We show that convention formation for efficient communication is a general challenge to state-of-the-art LLMs. Our post-training method effectively exploits the naturally occurring examples of conventionalization from human data. We observe significant improvements in model behavior, as reflected in evaluation scenarios that are distinct from the training data. We hope our work will encourage further research to improve LLMs' adaptation for communication efficiency.

Our work leaves various open problems, and points to potential directions for future work. We experiment with our approach in isolation. An important future direction is to integrate this intervention into the complete LLM post-training process, so that convention formation is acquired alongside the general skills acquired in post-training. Future work can also study the impact of this change on the broader set of post-training goals.

Critically, it is important to consider the incentives of human speakers when they form conventions: reduction of costs and increasing of utility. Models have no awareness of these considerations, and instead they learn the data distribution better through our process. Reasoning about cost and utility explicitly is an interesting direction for future work.

While our tasks avoid the potential issues of visual stimuli, they are intended to complement, not to replace, image reference games. Image referents in prior work cover diverse domains and levels of details (e.g., Hua & Artzi, 2024; Eliav et al., 2023), likely influencing language styles and adaptation. Future work can study the behavior differences under these settings.

Despite the significant improvements we demonstrate, our post-trained LLMs still do not show the same level of convention formation as humans. Closing the gap further is an important future work direction. Our evaluation benchmarks will enable this effort.

Finally, we focus on a special type of adaptation (i.e., convention formation). This is not the only way interlocutors adapt ad-hoc to make interactions more efficient and successful. For example, a speaker may sometimes want to switch (back) to more elaborate language, upon seeing the listener's confusion. We hope future work expands upon this set of behaviors.

## Acknowledgments

This research was supported by NSF under grants No. 1750499, a gift from Open Philanthropy, a gift from Apple, an Nvidia Academic Grant, the National Artificial Intelligence Research Resource (NAIRR) Pilot, the Frontera supercomputer supported by the National Science Foundation (award NSF-OAC 1818253) at the Texas Advanced Computing Center (TACC) at The University of Texas at Austin, and the Delta advanced computing and data resource, which is supported by the National Science Foundation (award NSF-OAC 2005572). We gratefully acknowledge use of the research computing resources of the Empire AI Consortium, Inc, with support from the State of New York, the Simons Foundation, and the Secunda Family Foundation (Bloom et al., 2025). We thank Robert Hawkins and Marten van Schijndel for insightful discussions. We thank the anonymous reviewers and the area chair for their valuable feedback.

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

## A   Human Study Details

To identify the group of people who likely misinterpreted the task, we calculate the average WNDs from Repetitions 4-6, where we expect conventions to have likely formed, and apply Gaussian Mixture Models (GMM) on the distribution of the WNDs for the 200 interactions collected. We tested component numbers from 1 to 8 and find that a 3-component GMM best fits the data according to the Bayesian Information Criterion. Based on this GMM (Figure 3), we identify the low-consistency group (Group 3 in Figure 3), which corresponds to 35.5% of the interactions. As shown by Figure 4, this group (purple curve) is overly verbose and achieves lower accuracy than the remaining interactions (gray curve) even though they start with about the same accuracy. Therefore, we only use the performance of the remaining interactions (Groups 1 and 2) as our goal and baseline for post-training.

In addition to the human-GPT interactions, we conducted a pilot study with 49 interactions where both the speaker and the listener were humans. We use the same GMM obtained from our 200 human-GPT interactions to divide the pilot study data into two groups. The low-consistency group corresponds to 30.6% of the pilot study data. We observe that the high-consistency groups from our human-GPT4o study and human-human study show a high agreement in message lengths, WND, and accuracies (Figure 4). Therefore, we conclude that GPT4o-mini is a valid substitute for human listeners to automate the reference game evaluation.

Distribution of Human Speaker WND Modeled by Gaussian Mixture Model

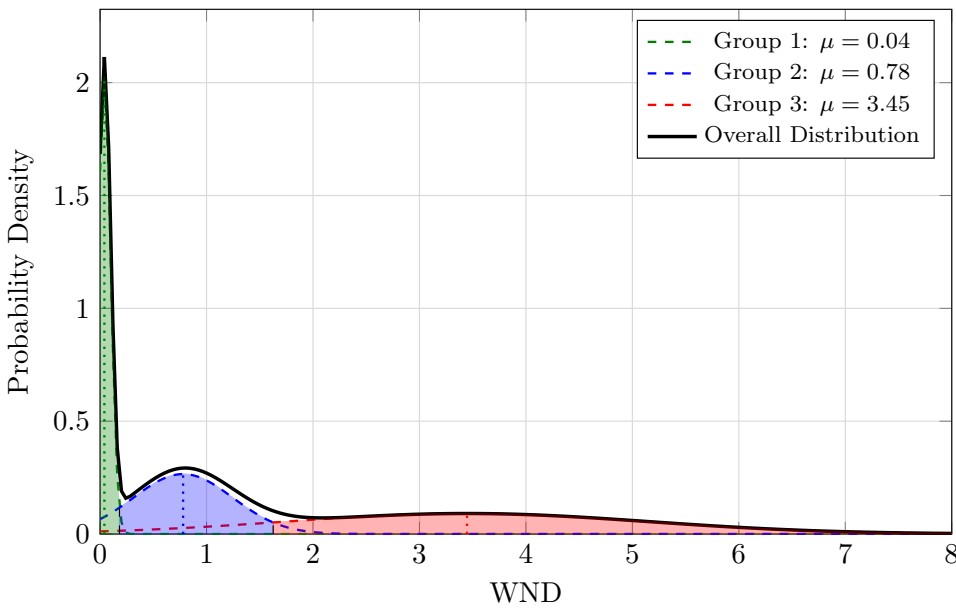

Figure 3: Distribution of Average Human Speaker WND in the second half of the interaction (Repetitions 4-6). Group 1 and Group 2 correspond to the convention-forming majority group which we use as the goal for our post-training.

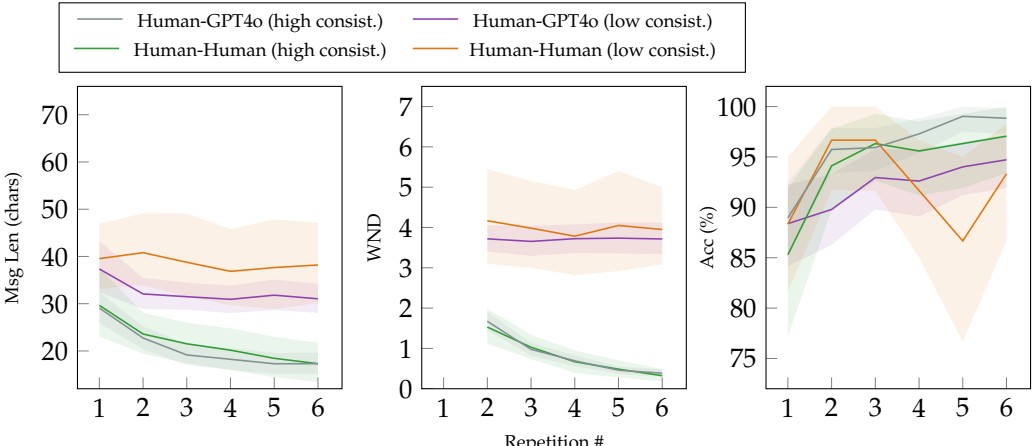

Figure 4: Comparison of our human-GPT4 interactions and our human-human pilot study. Shaded areas are bootstrapped 95% confidence intervals.

## B  Additional Implementation Details

**Training**   For both the SFT and APO stages, we search over three learning rates (5e-5, 1e-5, 5e-6) and use a linear decay learning rate scheduler. For SFT, we train for up to 300 updates with a batch size of 16. For APO, we use a batch size of 32 and experiment with two max number of updates (4000 steps and 10000 steps). For the SFT training to learn planning tokens, we set the $\omega$ in Equation 1 to 20 to rescale the JSD term to about the same order of magnitude as the cross-entropy loss. We use LoRA for all model training, using a LoRA rank of 16 for SFT and 32 for APO. We set LoRA alpha to 32 for both stages. For APO, we use the default beta of 0.1.

**Reference Games Context Construction**   To make the task sufficiently challenging, we construct referential contexts with semantically similar items. We use the OpenAI text-embedding-3-small model to generate item embeddings and follow the practice of Hawkins et al. (2020b) to use k-means clustering and k-nearest neighbors to extract contexts.

**Constraining the Generation Trajectory for Document-grounded Evaluation**   When given the references and past utterances, a model may choose to generate any appropriate response and sometimes does not re-mention any concepts from the context. We mitigate this with prefix constrained decoding. This does not guarantee a reference, but increases its likelihood by keeping the model roughly on the intended trajectory. Let $y$ be the original human utterance containing the re-mention and $y_p$ be the first N tokens before the re-mention we identified, we append $y_p$ to the model input such that the model is completing a sentence that already has the first N tokens filled. This constrains the potential trajectory of the model's text generation, so the model is more likely to refer to the same concept as the original human utterance. We use this input style for any test instances where the re-mention is not in the beginning of the original human utterance.

**GPT4o Judge for Document-grounded Evaluation**   To avoid assigning winning and losing labels due to trivial differences, we post-process the GPT4o judge's extracted re-mentions from the two candidate completions and compare the re-mentions' lengths. If the length difference is within 2 characters but GPT considers one candidate is winning, we override GPT's judgment and assign a label of tie.

To determine the reliability of the GPT4o judge, we manually annotate 200 instances for Gemma (original) vs. human completions, and calculate Krippendorff alpha and Cohen's kappa to reflect the agreement between our annotation and GPT4o's output. For Krippen-dorff alpha, we consider 'tie' and 'cannot decide' as the same category, so we calculate alpha for three labels ('win', 'lose', 'tie or cannot decide'). We use Cohen's kappa for our

competence rating (same definition from Section 6), where a competent model completion either wins over or ties with the human completion. We achieve a Krippendorff alpha of 0.705, greater than the 0.667 threshold to be acceptable for drawing tentative conclusions (Krippendorff, 2006). For the competence rating, our Cohen's kappa is 0.749, greater than the 0.6 threshold and indicates substantial inter-annotation agreement (Artstein & Poesio, 2008).

## C    Ablation Results

We present the reference game ablation results in Figure 5 (Gemma) and Figure 6 (Llama). We present the document-grounded evaluation results in Table 5.

We observe that using SFT only does not introduce the desired trends, illustrating the importance of preference pairs. It reaches near zero WNDs, because the model simply copies its first expression again and again, showing no efficiency improvement. The APO only baseline does bring significant improvement on the utterance completion task but not on reference games. Compared with our main method, APO only either increases the message lengths (for Gemma) or worsens message consistency (Repetition 6 WND=1.5 for Llama), thus not showing convention formation in reference games. It also causes degradation in reference game accuracies by up to 8% (for Gemma). These results show that the convention formation ability learned through APO only is not robust enough to generalize to an unseen scenario. Finally, the ablations on our method (w/o [remention] and w/o JSD) show less improvement on the utterance completion task and worse performance on reference games. We note that Llama w/o JSD is the only case where another post-training method produces overall shorter messages than our method. However, Llama w/o JSD does not show any length change and almost repeats the messages in exact words throughout the interaction (almost zero WND starting from Repetition 2). Additionally, this method causes lower reference game accuracies and leads to a smaller competence rate than our method.

These results indicate that both the SFT initialization with JSD regularization and the use of planning tokens are necessary components of our post-training pipeline, to bring generalizable convention formation abilities and balanced improvements across the tasks.

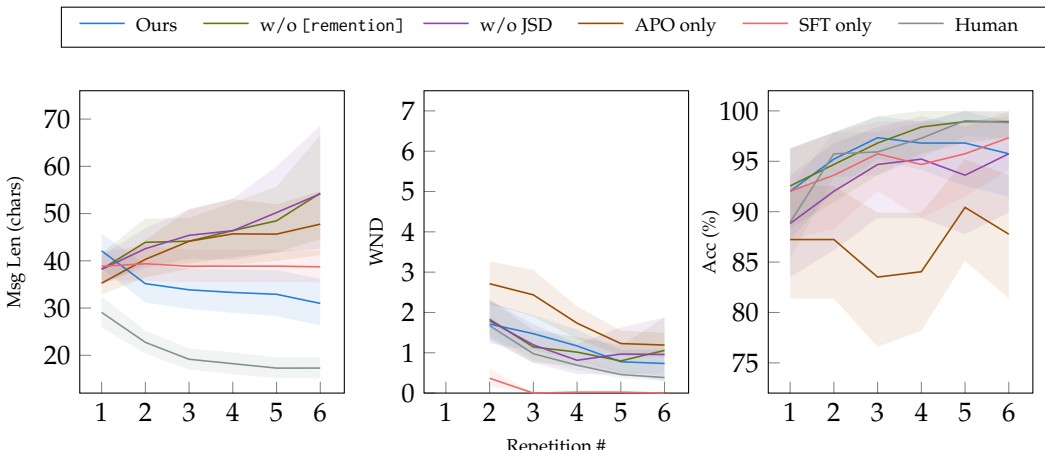

Figure 5: Reference game performance for ablations with Gemma. Shaded areas are bootstrapped 95% confidence intervals.

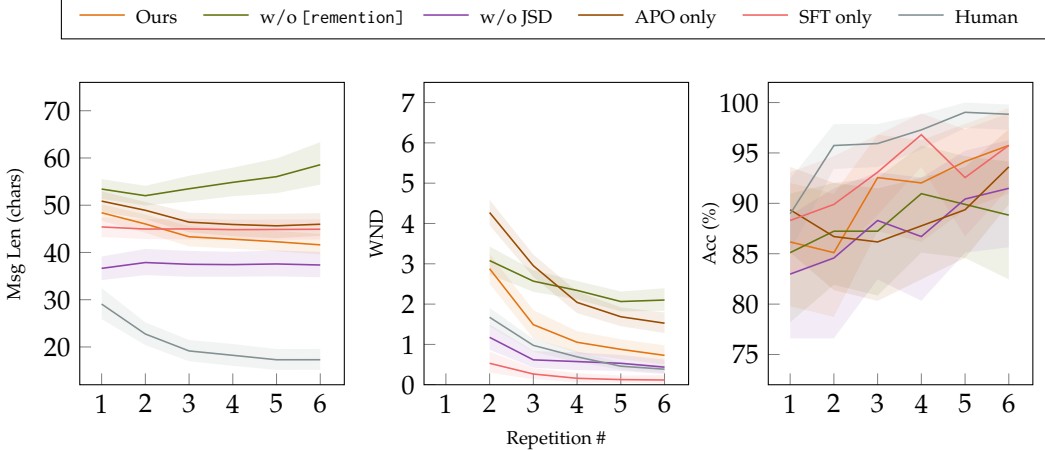

Figure 6: Reference game performance for ablations with Llama. Shaded areas are bootstrapped 95% confidence intervals.

| Method | # Wins | | | | Competence |
|---|---|---|---|---|---|
| **On Gemma** | LLM | Human | Tie | Cannot Decide | Compet. Rate (%) |
| Ours | 109 | 97 | 259 | 16 | 79.1 |
|    w/o [remention] | 104 | 115 | 239 | 23 | 74.9 |
|    w/o JSD | 100 | 115 | 253 | 13 | 75.4 |
| APO only | 126 | 89 | 242 | 24 | 80.5 |
| SFT only | 80 | 150 | 238 | 13 | 67.9 |
| **On Llama** | | | | | |
| Ours | 90 | 114 | 257 | 20 | 75.3 |
|    w/o [remention] | 85 | 130 | 244 | 22 | 71.7 |
|    w/o JSD | 100 | 130 | 234 | 17 | 72.0 |
| APO only | 99 | 117 | 246 | 19 | 74.7 |
| SFT only | 75 | 166 | 224 | 16 | 64.3 |

Table 5: Document-grounded utterance completion results for ablation studies

# D   Prompt Engineering For Convention Formation

Figure 7 presents the results for our various prompting-based attempts. In addition to the metrics we have introduced, we also report Word Novelty Rate (WNR) (Hua & Artzi, 2024), a length-normalized version of WND, which can more easily distinguish models' differences when some models' messages are extremely short (e.g., just 1 word).[6]

We first experiment with a prompt based on the Gricean quantity maxim, where we specify that the message should contain enough information for identifying the target but every token generated has a cost so the model should use a shorter message if it's enough. As Figure 7 ("Gricean") shows, under the Gricean prompt, the average shortening from the first to the last repetitions of the reference game was negligible for off-the-shelf models (dotted lines). We then experimented with adding an additional hint to the Gricean instruction: *think about how the amount of information needed may change as more rounds are completed and based on the listener's performance in previous rounds.* With this additional hint specific to reference games, we observe various extents of shortening on the original Gemma,

---

[6]In the extreme case where the messages just have 1 word and the model uses a new word every time, the WND will be at most 1 but there is no consistency. Although a model with good consistency will still have significantly lower WND (close to 0), its difference with the concise, inconsistent model may be hard to visualize (e.g., when the plot's Y-axis range needs to be much greater than 1 to accommodate for the larger WND values from earlier repetitions and/or other models).

Llama, and Claude, but they all show significantly worse consistency than humans, as shown by their higher WND and WNR (Figure 7 "Gricean+Hint"). This lack of consistency undermines communication efficiency, increasing the cognitive load for the partner to process the message and prevents convention formation (Hua & Artzi, 2024).

We also experiment with a more explicit instruction, where the prompt states, *as more rounds are completed and the listener understands you better, make your referring expressions shorter and shorter every round*. This brings significant shortening, but three of the four models tend to produce inconsistent messages (Gemma, GPT, and Claude), as shown by their significantly higher WND and WNRs than human's in Figure 7, "Explicit". This undesirable behavior of substantial shortening without consistency is also observed in image reference games, as reported by Hua & Artzi (2024). We find that only very explicit instructions on how to shorten the messages can bring both consistency and shortening (Figure 7, "Explicit + Consistency"). We have to explicitly instruct the model to *extract salient tokens from the previous messages for this item rather than introducing new words... For each item, when you reach a message you think can not be further shortened ..., you should keep using that message for the rest of the game*. Such heavy-handed prompts will not generalize beyond reference games, so prompting is not a feasible solution to elicit convention formation.

We additionally experiment with in-context examples to explain convention formation. Our prompt first states that *convention formation happens when people's referring expressions for an item evolve and become more efficient over time*. It then provides examples showing how the expressions for the same item become more efficient over time (Figure 8). This combination of a general explanation of convention formation and in-context examples is more likely to generalize than the explicit, reference-game-specific instructions tested above. Therefore, we apply this prompt to the document-grounded evaluation as well.

Under this few-shot prompt, we still observe that all models (except for Claude) produce significantly less consistent messages than humans in reference games (as shown by their higher WND and WNRs in Figure 7, "Few-Shot"), thus not showing convention formation. Besides the inconsistency issue, we find that the shortening brought by the few-shot prompt does not generalize well to the document-grounded evaluation. The few-shot prompt only brings small improvements in competence rates: Gemma (71.9 $\rightarrow$ 73.9), Llama (66.0 $\rightarrow$ 66.9), Claude (63.8 $\rightarrow$ 68.2), and GPT (59.0 $\rightarrow$ 62.7). All of these improved scores are still substantially worse than what our post-trained models achieve under just the standard prompt (Gemma (ours): 79.1, Llama (ours): 75.3). Therefore, we conclude that prompting, even with in-context examples, is not a shortcut to elicit convention formation in LLMs.

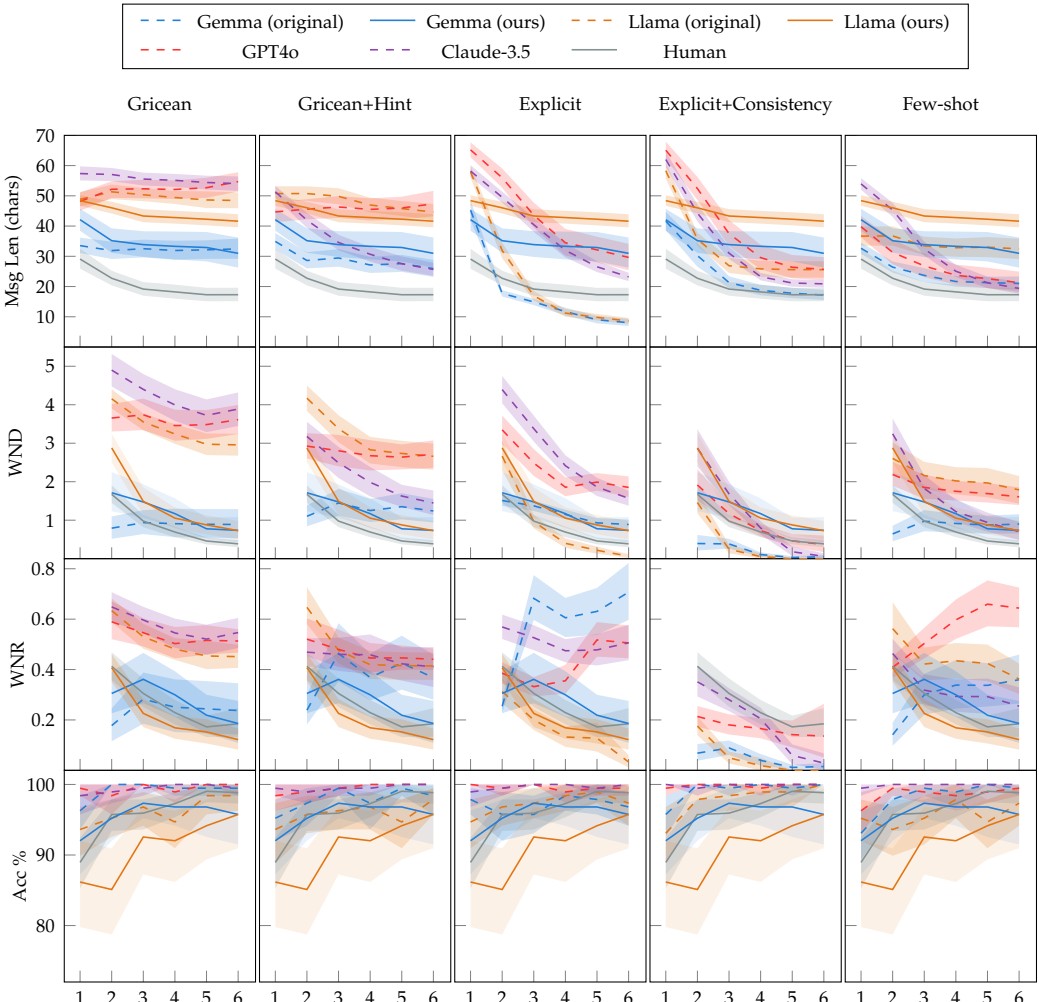

Figure 7: Attempts to elicit convention formation with prompting. We only use the standard prompt with our post-trained models. Shaded areas are bootstrapped 95% confidence intervals.

[System] You will speak naturally just like a human and, when applicable, you should show a behavior called convention formation. Convention formation happens when people's referring expressions for an item evolve and become more efficient over time. Each of the examples below shows how the referring expressions for an item may evolve when it's mentioned for the first time, second time, third time, and so on.
Example 1:
1) the ceramic vase with floral patterns
2) the floral ceramic vase
3) the ceramic one
4) the ceramic

Example 2:
1) a bowl full of mixed fruit with black background
2) the mixed fruits, black background
3) the mixed fruits, black background
4) the black background

Example 3:
1) a building with a small fire hydrant in front of a big pole
2) the building with the fire hydrant
3) the fire hydrant
4) the fire hydrant

[... Reference Game Instructions ...]

Figure 8: Few-shot prompt to elicit convention formation.

# E Examples

## E.1 Example Inputs

---

Write the next line of this excerpt of TV show transcript from where it's left off. You will play the role of Frasier.
[Scene Three - Radio Station. Frasier is rounding up his show as Roz reads Dr. Mary's book in her booth.]
Frasier: Well, Seattle, thank you for your calls.
[Frasier knocks on the window to Roz who is immersed in [remention] the book. She carries on reading.]
Frasier: Seattle, thank you for your calls!
Roz: [monotone and bored, still reading] Hey, Frasier, what are you doing over the Christmas weekend?
Frasier: Well, Roz, if you insist on interrogating me, I'll be co-hosting the Seattle Christmas parade tomorrow night...
[Roz begins nonchalantly ringing some Christmas bells whilst still reading. She is obviously not too fussed about Frasier's plans for finishing the show.]
Assistant (Frasier):

[Preferred completion $y_w$]: ... I hope it will be the beginning of a new holiday tradition. Good mental health, see you at [remention] the parade.

[Dispreferred completion $y_l$]: ... I hope it will be the beginning of a new holiday tradition. Good mental health, see you at [remention] the Seattle Christmas parade.

---

Figure 9: Example training data

[System] Complete a repeated reference task with a listener. You will act as the speaker and I will be the facilitator. This task consists of multiple rounds in which you interact with the listener on the same referential context (represented by 4 word cards). In each round, I give you one of the 4 words as the target. You should refer to the target with a referring expression describing the target, without mentioning the target word itself or any words in the referential contexts. If any of the context words is a compound word, you cannot mention their components or any part of the word either. The listener will try to choose the target correctly based on your referring expression. The listener will see the 4 words in a different order every round so you cannot refer to the target by saying where it is in the context. You can use whatever description you think will be helpful to the listener. Make sure your expression can distinguish the target from other items. The same item will appear as the target multiple times throughout the task, so you will repeatedly refer to what has appeared before. You can use up to 15 words for your referring expression. You can only see the word cards representing the items, not the actual objects. Reply by telling the listener: Please pick [how you refer to the target].

Referential Context: "cleaning bucket", "kitchen broom", "laundry basket", "dustpan"

[Trial 1] Target: "laundry basket"
[Speaker] Please pick the big, open container with the handle.
[System] The listener mistakenly answered "cleaning bucket".

[Trial 2] Target: "cleaning bucket"
[Speaker] Please pick the yellow container with the handle and the blue sponge inside.
[System] The listener answered correctly.

[Trial 3] Target: "dustpan"
[Speaker] Please pick the flat, rectangular thing with the handle that you sweep into.
[System] The listener answered correctly.

...

[Trial 24] Target: "laundry basket"
[Speaker]

Figure 10: Example model query for the text-only reference game.

[Reference Document]
"Beginning Summer 2018, there will be new rules regarding dealer record keeping. [1] To register as a retail dealer licensed by the Department of Motor Vehicles in New York State consult the DMV Commissioner s Regulations for Motor Vehicle Dealers and Transporters ...
DMV does not provide official dealer signs. You may purchase these signs from various sign shops and automotive supply stores. Your sign needs red background and white lettering to measure at least 3 feet wide by 2 feet tall display REGISTERED Facility Number STATE OF NEW YORK MOTOR VEHICLE DEALER in two inch block lettering, ..."

[Reference Span]: "Your sign needs red background and white lettering to measure at least 3 feet wide by 2 feet tall"

Generate the next utterance based on the reference document and the specific excerpt of the conversation (the "reference span"). The reference span likely has information about the solution to the user's query/request.

[Conversation]
User: We want to open a snowmobile dealership
Assistant: Would you like to know how to apply for a snowmobile dealer business certificate?
User: Yeah.
......
User: Last question, what does our official registered dealer sign have to be size-wise?
Assistant:

Figure 11: Example model query for the document-grounded utterance completion task. A good model should refer to the "official registered dealer sign" with a more concise phrase.

[Context]
User: We want to open a snowmobile dealership
Assistant: Would you like to know how to apply for a snowmobile dealer business certificate?
User: Yeah.
......

User: Last question, what does our official registered dealer sign have to be size-wise?
Assistant:

[Completion A] Your sign needs to be at least 3 feet wide by 2 feet tall . . .

[Completion B] Your official registered dealer sign needs to be at least 3 feet . . .

Above are two candidate completions for the agent's turn. Which completion shows stronger convention formation? Convention formation manifests when the referring expression for an item or a group of items becomes more concise after the item was initially mentioned. For example, an item initially mentioned as 'the slim ceramic vase that has floral patterns' may later be re-mentioned as 'the floral ceramic vase' or simply 'the ceramic vase.'

Consider the items in the [context] that are re-mentioned in both completions. The completion that uses more concise phrases (shorter phrases) overall when re-mentioning those items is considered showing stronger convention formation. Remember that a re-mention is the exact noun phrase referring to the item, which includes all the words, phrases, and clauses modifying the noun. These modifiers will also affect the verbosity (conciseness) of the re-mention. Also, the same item may be re-mentioned multiple times in a completion and the two completions may eventually reach the same level of conciseness for that item. In this case, one completion may still show stronger convention formation by using the more concise referring expression more often.

Your answer should follow one of the following formats: 1) If one completion shows stronger convention formation overall, output its label (A or B) and an example (evidence) where an item is re-mentioned more concisely in that completion than in the other.
2) If the completions show the same level of convention formation, output C and an example (evidence) where the re-mentions in the two completions have similar conciseness.
3) If you cannot find any example (evidence) to make a judgment, output D.

Return a json of the following format: {"initial mention": "", "re-mention in Completion A": "", "re-mention in Completion B": "", "stronger convention formation in": "<A, B, C, or D>"}.

Figure 12: Example query for GPT4o judge in document-grounded utterance completion.

## E.2 Interaction Examples

We present positive examples and various negative examples in the following figures. Each example represents an interaction and shows how the messages for each item evolve across 6 repetitions. We highlight in red the messages that led to the listener's mistake. If a message is not highlighted in red, it means the listener correctly identified the target when seeing the message. Unless otherwise specified, all the negative examples represent issues found across the two model families (Gemma and Llama) and in both post-trained models and their off-the-shelf versions.

Figure 13 and Figure 14 present positive examples for convention formation. From Repetition 1 to 6, the messages show substantial shortening overall, and the model rarely introduces new words to refer to the same item. The listener consistently selects the target correctly.

Among the negative examples, the most conspicuous type is where the model does not show any shortening and just repeats the initial messages (Figure 15). Another important type of failure found in all models is using a message that is relevant but not specific enough (Figure 16). We also found issues only observed in some models. The first one, found in Llama (both post-trained and original) and Gemma (post-trained), is that the model occasionally produces a message that refers to another item in the context rather than the target (Figure 18). The second one is specific to the post-trained Gemma, where the model occasionally hallucinates the location of the target items (Figure 19).

| Bowl |
| --- |
| 1. The round, white thing that is bigger than the plates but smaller than the pot |
| 2. The round thing that is bigger than the plates but smaller than the pot |
| 3. The round thing that is bigger than the plates but smaller than the pot |
| 4. The round thing |
| 5. The round thing |
| 6. The round thing |

| Butter Dish |
| --- |
| 1. The small, square, white thing that holds the yellow stuff |
| 2. The square, white thing that holds the yellow food |
| 3. The square, white thing that holds the yellow food |
| 4. The square, white thing |
| 5. The square thing |
| 6. The square thing that holds the yellow food |

| Dishes |
| --- |
| 1. The set of plates and cups |
| 2. The plates and cups |
| 3. The plates and cups |
| 4. The plates and cups |
| 5. The plates and cups |
| 6. The plates and cups |

| Kettle |
| --- |
| 1. The metal thing with the handle that you put water in |
| 2. The metal pot with the handle |
| 3. The pot that you put water in |
| 4. The pot |
| 5. The pot with the handle |
| 6. The pot |

Figure 13: Example showing improved efficiency via convention formation. Model: Gemma (Ours).

**Coffee Filter**

1. The thing you use to separate liquid from solid particles
2. The thin material that holds back small particles
3. The thin thing that catches small particles
4. The thin thing that holds back particles
5. The thin thing that catches particles
6. The thin thing that catches particles

**Coffee Machine**

1. The device that makes a hot drink in your kitchen
2. The device that makes a drink from a bean
3. The device that makes a hot drink
4. The device that makes a hot drink
5. The device that makes a hot drink
6. The device that makes a hot drink

**Tea Pot**

1. The object you use to make a hot drink
2. The object that holds a hot liquid
3. The object that holds a hot drink
4. The object that holds a liquid
5. The object that holds a hot liquid
6. The object that holds a liquid

**Washing Machine**

1. The device that cleans clothes in your home
2. The appliance that uses water to clean clothes
3. The device that cleans clothes with water
4. The device that cleans clothes in water
5. The device that cleans clothes in water
6. The device that cleans clothes

Figure 14: Example showing improved efficiency via convention formation. Model: Llama (Ours).

**Bowl**

1. The round, open container, bigger than a cup
2. The round, open container, bigger than a cup
3. The round, open container, bigger than a cup
4. The round, open container, bigger than a cup
5. The round, open container, bigger than a cup
6. The round, open container, bigger than a cup

**Butter Dish**

1. The small container with a lid, used for holding spread
2. The small container with a lid, used for holding spread
3. The small container with a lid, used for holding spread
4. The small container with a lid, used for holding spread
5. The small container with a lid, used for holding spread
6. The small container with a lid, used for holding spread

**Dishes**

1. The set of plates and cups
2. The set of plates and cups
3. The set of plates and cups
4. The set of plates and cups
5. The set of plates and cups
6. The set of plates and cups

**Kettle**

1. The thing you use to boil water
2. The thing you use to boil water
3. The metal container with a spout, used for boiling water
4. The metal container with a spout, used for boiling water
5. The thing you use to boil water
6. The metal container with a spout, used for boiling water

Figure 15: Example showing repetition and no convention formation. Model: Gemma (Ours).

**Laundry Basket**

1. The big, open container with the handle ✗
2. The big, open container with the handle
3. The big, open container
4. The container with the handle
5. The container with the handle
6. The container with the handle

**Cleaning Bucket**

1. The yellow container with the handle and the blue sponge inside
2. The yellow container with the sponge
3. The yellow container
4. The yellow container with the sponge
5. The yellow container
6. The yellow container

**Dustpan**

1. The flat, rectangular thing with the handle that you sweep into
2. The rectangular thing with the handle
3. The thing you sweep into
4. The rectangular thing
5. The rectangular thing with the handle
6. The rectangular thing

**Kitchen Broom**

1. The long, thin thing with the bristles on the end
2. The brown, long thing with the bristles
3. The brown, long thing with the bristles
4. The brown, long thing
5. The brown, long thing with the bristles
6. The brown, long thing

Figure 16: Example interaction that started with an ambiguous referring expression. The messages still evolved as the game progressed. Model: Gemma (Ours).

**Light Bulb**

1. The thing you turn on in a room to see in the dark
2. The thing that glows when you turn on a switch
3. The thing that is often used in a lamp
4. The thing that is often used in a ceiling fixture
5. The thing that is used in a socket
6. The thing that is used in a lamp

**Satellite**

1. The object that orbits the earth and sends signals back to it
2. The object that you can see in the sky at night
3. The object that is launched into space from a rocket
4. The object that is used for navigation
5. The object that is launched from a rocket
6. The object that is in orbit around the earth

**Smartphone**

1. The device you use to talk to people and play games
2. The device you use to watch videos and listen to music
3. The device that you use to take pictures and videos
4. The device that you use to access the internet
5. The device that you use to make calls
6. The device that you use to watch videos

**Solar Cell**

1. The device that converts energy from the sun into electricity
2. The thing that powers your phone and other small devices
3. The thing that is used to charge your phone
4. The thing that is used to power a house
5. The thing that is used to charge a battery
6. The thing that is used to power a house

Figure 17: Example showing inconsistent messages and no convention formation. Model: Llama (Ours).

**Steam Iron**

1. The thing you hang your shirts on to dry ✗
2. The tool you use to make your shirt smooth
3. The tool you use to get rid of wrinkles
4. The tool you use to make your shirt smooth
5. The tool you use to make your shirt smooth
6. The tool you use to make your shirt smooth

**Cutting Board**

1. The flat surface you chop food on in the kitchen
2. The thing you chop food on in the kitchen
3. The surface you chop food on in the kitchen
4. The thing you chop food on in the kitchen
5. The thing you chop food on in the kitchen
6. The thing you chop food on in the kitchen

**Ironing Board**

1. The thing you use to get wrinkles out of your shirt ✗
2. The flat surface you put your hot tool on
3. The flat surface you put your hot tool on
4. The flat surface you put your hot tool on
5. The flat surface you put your hot tool on
6. The flat surface you put your hot tool on

**Clothesline**

1. The long string you hang your wet laundry on outside
2. The thing you hang your dirty laundry on to dry
3. The thing you hang your laundry on outside
4. The thing you hang your laundry on outside
5. The thing you hang your laundry on outside
6. The thing you hang your laundry on outside

Figure 18: Example showing the model erroneously referring to a non-target. The first message for the steam iron is erroneously referring to the clothesline. The first message for the ironing board is also more suitable for the steam iron. Model: Llama (ours).

**Gas Turbine**

1. The thing that uses a spinning wheel to produce power
2. The thing that uses a combustion engine
3. The thing that is in the power plant ✗
4. The thing that is the second biggest
5. The thing that is next to the pipes ✗
6. The thing that is in front of the control panel

**Light Bulb**

1. The thing that produces a bright white glow
2. The thing that is screwed into a socket
3. The thing that is in the lamp
4. The thing that is in the ceiling
5. The thing that is in the middle ✗
6. The thing that is in front of the pipes ✗

**Washer Dryer**

1. The appliance that cleans clothes
2. The appliance that is in the laundry room
3. The appliance that is next to the sink
4. The appliance that is on the floor
5. The thing that is on the left ✗
6. The thing that is on the right ✗

**Electric Generator**

1. The thing that produces power for the other items
2. The thing that is connected to the power grid
3. The thing that is next to the transformer
4. The thing that is the biggest ✗
5. The thing that is next to the control panel
6. The thing that is behind the pipes ✗

Figure 19: Example showing the model hallucinating the target's location/position. The model hallucinates locations such as *next to the pipes*, *on the left*, and *in the middle*, which confuse the listener. Model: Gemma (Ours).

