# OpenReview forum: "Post-training for Efficient Communication via Convention Formation"
_colmweb.org/COLM/2025/Conference — COLM 2025_

### Official Review · Reviewer_Usy9 · 2025-05-05

**Rating:** 6
**Confidence:** 3
**Ethics Flag:** 1

**Summary:**

This paper examines LLMs' capability of convention formation (CF) -- whether LLMs can learn to use simplified mentions to facilitate efficient multi-turn interactions. The authors propose a DPO-based pairwise training algorithm to improve CF for LLMs by encouraging LLMs to generate simplified mentions while discouraging the exact copy. They evaluate CF using both synthetic reference games (measuring utterance length and word distance) and a realistic conversational document QA benchmark, judged by GPT-4o. Results show that while off-the-shelf LLMs struggle to adopt simplified mentions and enhance conversational efficiency, the proposed training method effectively addresses this limitation. Ablation studies further validate the contribution of each algorithmic component. Further ablation study confirms the effectiveness of each component in the algorithm.

**Questions To Authors:**

1. It is not that clear to me what is the "Accuracy" of text-only reference games (Figure 2, the third subfigure) -- what does it measure, and why it is relevant?

2. Though perhaps not in a multi-round interaction setting, there should exist many evaluation environments related to reference games and featuring speaker-listener setup in LLM/NLP literature (e.g., theory-of-mind, pragmatics). What would be the reason that we need to build a new reference game setup? And what would be the main difference between the proposed evaluation tasks to existing related textual tasks?

**Reasons To Accept:**

1. The proposed convention formation task provides a novel interesting angle to evaluate LLM multi-turn interaction capability. The evaluation setup for both synthetic games and realistic applications is well-considered.

2. The proposed DPO-based training method, while simple, yields significant improvements over baselines.

**Reasons To Reject:**

1. **Debatable formulation**: The central assumption—that using simplified re-mentions improves multi-turn interaction efficiency—may not universally hold. In cases where acronyms are introduced abruptly, listeners may lack sufficient context to interpret them correctly. Similarly, in formal or high-stakes scenarios, omitting parts of important entity mentions could increase the risk of misinterpretation, potentially undermining clarity and hurting the interaction efficiency. There should be more discussions related to this problem.

2. **Weak baselines**: To establish the necessity of training to improve models' capability of handling CF, the paper should consider stronger inference-time baselines. For instance, can we mitigate CF problems by providing just a few in-context samples in the curated datasets or by carefully designed system prompts for LLMs? LLMs are not explicitly trained to do well on the proposed task, and it is thus expected direct prompting might lead to performance degradation, but with a few prompting tweaks it might work.

3. **Lack of in-depth analysis**: Although the proposed methods show promise based on automatic metrics, it is not clear what happened behind those performance numbers. For example, what would be common strategies that LLMs learn to reduce message length? Do they favor pronouns, drop adjectives, invent acronyms, or employ other tactics?  More in-depth qualitative or quantitative analysis would further strengthen this paper.

---

> ### Author Response · Authors · 2025-05-31
> **Author Response**
>
> Thank you for your comments and questions!
>
> **“The central assumption—that using simplified re-mentions improves multi-turn interaction efficiency—may not universally hold.”**
> This is a great point and we completely agree that the behavior humans display is complex, and takes into account various considerations. It is important to note that we are not making this assumption though. The key of our formulation is that the model should show convention formation whenever humans do, and we design our baselines and evaluations closely around human behaviors to make sure that the convention formation behaviors tested indeed manifest improved communication efficiency. The human distribution of convention formation implicitly accounts for the important consideration you raise. We will refine our discussion to make this clear.
>
> **“The paper should consider stronger inference-time baselines”**
> We conducted pilot early pilot studies with in-context samples but did not observe significant improvements. It’s also not clear what ICL examples to use, because we can’t just specialize them for the evaluation tasks, because that won’t be a general solution. In our post-training method, we intentionally designed the method to use a data domain and format that differ from the evaluation tasks, so that we are more confident about the generalizability of the evaluation results (we never train the model on the evaluation tasks). Tweaking the prompts and few-shot examples would require using the evaluation tasks directly to improve performance metrics. This will further hurt any chance of fair comparison with our post-training approach. The more we tweak the prompts based on the evaluation tasks, the more likely we make the prompts specific to the evaluation tasks and not generalizable. It may be the case that an extensive study of how to create such behaviors with ICL examples will succeed in eliciting general behaviors, but this is really an open problem, and we consider it an alternative approach and not merely another baseline.
>
> **“Lack of in-depth analysis… Do they favor pronouns, drop adjectives, invent acronyms, or employ other tactics? More in-depth qualitative or quantitative analysis would further strengthen this paper.”**
>
> These are great suggestions. We will add these analyses to the next version of the paper. Our submitted draft did not include these analyses because the paper was already dense in content under the 9-page limit given that our contributions include both evaluation tasks and post-training methods.
>
> **“What is the "Accuracy" of text-only reference games?”**
> The “accuracy” in text-only reference games refers to the accuracy of the listener for selecting the correct target item. It’s the same metric used in the conventional, image-only reference games from the prior works (e.g., [Hua and Artzi, 2024](https://arxiv.org/pdf/2408.01417); [Hawkins et al., 2020b](https://aclanthology.org/2020.conll-1.33.pdf)).
>
> **“What would be the reason that we need to build a new reference game setup? And what would be the main difference between the proposed evaluation tasks to existing related textual tasks?”**
>
> The text-only aspect of our reference game is essential for evaluating convention formation in LLMs that do not possess vision capabilities (which are a large proportion of contemporary LLMs). It also removes the potential confounding factor (in LLMs) caused by the visual stimuli in the conventional reference games from cognitive science. Our contribution includes introducing new interaction rules and codebase that makes the text-only games non-trivial and suitable for convention formation (e.g, the speaker cannot just simply copy a part of the target word and keep using it (Lines 190-193); the referential contexts are sufficiently challenging).
>
> The multi-turn aspect of our reference game is inherited from cognitive science studies and is essential for studying the key properties of convention formation and efficient communication. It allows us to study the consistency of referring expressions across multiple re-mentions and the improved accuracy of the listener as the two participants interact more.
>
> Our tasks create scenarios where a good communicator (e.g., a human) would adapt their language to be more efficient and introduce tools to quantify this behavior. Other textual tasks cannot fulfill this purpose.

---

> > ### Comment · Reviewer_Usy9 · 2025-06-01
> > **Thanks for the response!**
> >
> > I am grateful for the authors' quick responses to my reviews. These responses solve many of my questions. Just a few more comments to follow up:
> >
> > 1. Please make sure to include the refined discussions on research question formulation, and in-depth behavior analysis that LLMs did to improve communication efficiency in the revised version. In my mind, in-depth analysis is much more important than aggregated benchmark numbers because it provides more actionable insights than aggregated benchmark numbers.
> >
> > 2. Regarding ICL examples and generalizability: Please allow me to clarify a bit -- I do not require you to do extensive prompt optimization using the evaluation task. There should be some even simpler trials that hold across domains, for example writing some general system prompts ('Take care of your language and make it as concise as possible. Feel free to form new conventions to improve communication efficiency.") and then showing some cases on how human uses acronyms to simplify the conversation. I think it should not be hard to make up some toy examples that hold across domains.
> >
> >     My main concern with regard to a training-based solution is that, as I said in the first weakness point of my review, improving communication efficiency in the real world could be very complicated and situation-dependent. It is not sustainable to train a new model for a new situation, also training can bring unexpected performance damage to the model's capability. If we want to push a training-based method, regardless of this clear drawback, we should first know the limitation of straightforward inference-time interventions.
> >
> > 3. Regarding discussions on other textual datasets: Thanks for the explanation, but I am still not very clear when the author explained "Other textual tasks cannot fulfill this purpose" -- do you mean other textual datasets are not designed to fulfill the intended evaluation, or they cannot be **adapted** to do so? I can imagine a very toy scenario where you adapt some dialogue datasets and append some round counter/word counter to the feedback/response from the listener in some intermediate rounds, and a well-tuned speaker should be able to use this clue to do conversation efficiency optimization. An alternative approach, similar to the author's adaption on Doc2Dial, is to adapt the reward setting in BanditQA [1] to encourage convention formation explicitly. To be fair, I am not requesting the author to run new experiments. A detailed and thorough discussion of related works should suffice.
> >
> > References:
> >
> > [1] Gao, Ge, Eunsol Choi, and Yoav Artzi. "Simulating Bandit Learning from User Feedback for Extractive Question Answering." Proceedings of the 60th Annual Meeting of the Association for Computational Linguistics (Volume 1: Long Papers). 2022.

---

> > ### Author Response · Authors · 2025-06-03
> > **Thank you for the follow-up!**
> >
> > We appreciate the quick follow-up! We are happy to hear we addressed many of the questions. If our response solves the issues you raise, we will appreciate it if you reflect this in updating your rating, because the mean rating is a major factor in how decisions are made. Thank you\!
> >
> > **Response to Comment 1**
> > Yes, we will include all these items. We agree about the value of qualitative behavior analysis.
> >
> > **Response to Comment 2**
> > Thank you for your clarification\! We tested various in-context exemplars for few-shot learning but all the results were negative. There may be a smart way to select exemplars and generate prompts better, but this is outside the scope of this paper. We tested general system prompts similar to the style you suggested but did not observe positive results. We started with a prompt based on the Gricean quantity maxim, saying that the message should contain enough information to identify the target but every token has a cost so the model should use a shorter message if it’s enough. Under this prompt, the shortening between the first and the last repetitions of the reference game was still negligible (shortened by 1.1 characters for Gemma, 0.4 for Llama, 3.3 for Claude; the length increased by 6.5 for GPT).
> >
> > We also tried more explicit system prompts, such as “as more rounds are completed, make your referring expressions shorter and shorter every round,” but this led to poor message consistency for all models except for Llama. The WNDs at the end of each reference game were 1.85 (GPT), 1.57 (Claude), 0.89 (Gemma), and 0.06 (Llama). Importantly, the 0.89 WND for Gemma indicates poor consistency in this case because Gemma’s average message length under the explicit prompt reduces to 8.1 characters (about 1 content word) in Repetition 6, much shorter than the human average of 17.3 characters. Given that WND is the count of new content words not found in the previous mention, the WND of 0.89 here means most of the messages are inconsistent. This issue is not found in our post-trained models.
> >
> > We found that only very reference-game-specific and explicit prompts on how to shorten and maintain consistency can elicit convention formation without training, such as “extract salient tokens from the previous messages for this item rather than introducing new words… when you reach a message that can not be further shortened ..., you should keep using that message ...” These prompts are scenario-specific, and do not generalize. These observations closely track the findings on image reference games reported by Hua and Artzi, 2024, which suggests that prompting is not a general solution to convention formation and is the main motivation for our post-training work. We will elaborate on these negative results of prompting-based methods in our revised version.
> >
> > **Regarding your concerns on training-based solutions**: we explicitly design our framework to develop and evaluate convention-formation as a general capability that should manifest in different scenarios rather than following the paradigm of training a new model for a specific new scenario. This is why we made our training and evaluation scenarios completely different, and followed designs from Cognitive Science lab studies in evaluation.
> >
> > You raised a good point on unexpected damage to model capabilities brought by post-training. We agree that a solution should add the convention formation ability to a general-purpose LLM without hurting its existing capabilities. We design our approach to achieve this with careful regularization. It’s very challenging to test for such damage though, and most targeted post-training work doesn’t do it. We tested our models’ performance before and after post-training on the MixEval-hard benchmark, which covers a wide range of LLM tasks and has the state-of-the-art correlation with human judgment. Our trained models do not show clear performance degradation on this benchmark: for Gemma and Llama respectively, the accuracy-based MixEval score changes are merely \+0.2 and \-0.8 percentage points. On the other hand, prompting-based solutions to LLMs’ deficiencies can also cause unexpected performance damage. \[1\] shows a good example. We will add these explanations to our revised draft.
> >
> > \[1\] Shaikh, Omar, et al. "[Grounding Gaps in Language Model Generations](https://arxiv.org/pdf/2311.09144)"
> >
> > **[Response to Comment 3 continues in the next post]**

---

> > > ### Author Response · Authors · 2025-06-03
> > > **(Continues from the previous post) Response to Comment 3**
> > >
> > > Response to **Comment 3:**
> > >
> > > Our intention when writing "other textual tasks cannot fulfill this purpose" is to say that other textual tasks are not designed to fulfill this purpose and adapting them for evaluating convention formation is not trivial (i.e., requires substantial effort and extensive re-design). This is well illustrated by the Doc2dial dataset in our grounded utterance completion task. The original dataset is not directly usable for evaluating convention formation. It required significant work and thinking. For example, we used a coreference-based model to identify instances where convention should form and design test instances based on them. This is not to say that other datasets can’t be adapted, but just to say it’s not trivial.
> > >
> > > Regarding the suggested toy scenario, we want to clarify that for this paper, we want to give LLMs the convention formation ability without additional requirements on how they need to be used. In other words, the post-trained model should be able to seamlessly replace their off-the-shelf counterpart in production, without requiring changing the inference interface to provide additional clues for the model to use.
> > >
> > > Regarding the reward setting of BanditQA: it could be an alternative to our training algorithm (it does rely on human feedback, which entails a very different setup than the one we study). We will discuss alternatives to our current algorithms in our revised draft, including designing explicit reward functions for other RL methods.

---

> > > > ### Comment · Reviewer_Usy9 · 2025-06-08
> > > > **Thanks for the detailed comments!**
> > > >
> > > > I sincerely thank the authors for their detailed feedback. I am interested in the "negative results" that the authors obtained for "Response 2" -- I feel incorporating this analysis and observation, potentially with a bit more in-depth analysis on the suboptimal strategy that the model may use by simple prompting, would be much better motivate the proposed training methods. I would advise that authors reconsider the paper structure when incorporating these new contents. I raise my score a bit to show support for the paper's acceptance if all promised changes are implemented in the final revision.

---

> > ### Author Response · Authors · 2025-06-09
> > **Thank you and additional analyses on language behaviors**
> >
> > Thank you for raising your rating! We will implement all the changes we promised in the final revision.
> >
> > We also followed your suggestion and analyzed the model and human behaviors in reference games. Here’s an overview of the results:
> >
> > Humans and LLMs **did not favor using pronouns or acronyms**. Pronouns were not preferred likely because of their inherent ambiguity when the speaker needs to distinguish the target from other items. As for acronyms, we did not notice a clear instance of developing acronyms even in human messages. The only close example was when the human message for an item changed from “the one I don’t know” to “one idk”. We think developing acronyms would be a more common strategy in interactions that last for a longer time (e.g., multiple conversations over a period of time), as in some professional scenarios. Our work focuses on ad-hoc conventions within a single conversation. Prior human studies in Psychology did not report developing acronyms as a strategy in this scenario either \[1\].
> >
> > Instead, **the common strategies for both humans and models are dropping phrases, rephrasing, and describing a new feature**. We analyzed humans’ and Gemma’s language changes from one repetition to another for 30 reference games. Table 1 presents the results. The most common strategy is to remove a phrase from the previous message while keeping the rest of the message the same (“Drop Phrase”). The removed phrase is often a prepositional phrase or a clause. Another strategy is rephrasing, using a concise expression to replace part or all of a message while conveying a similar meaning. For example, the referring expression for a “waterbed” can change from “the mattress that is filled with liquid” to “the liquid-filled mattress”. Finally, both humans and Gemma sometimes replace the previous description by describing a new feature that is more concise. For example, a hammer can be initially described as “the tool that is used to fix things around the house” and later as “the tool that is used to build things”. These analyses show that humans and Gemma share similarities in their strategies, though Gemma’s more frequent use of new features likely has contributed to the slightly lower language consistency than humans (as our WND shows). We will add our analysis for Llama in our camera-ready draft as well.
> >
> > **Table 1**: Common Convention Formation Strategies
> >
> > |  | Drop Phrase | Rephrase | New Feature |
> > | :---- | ----- | ----- | ----- |
> > | Human | 66% | 19% | 15% |
> > | Gemma (ours) | 62% | 11% | 26% |
> >
> > Additionally, we counted the POS tags for models and humans, looking at how the counts change over time (Table 2). For all systems, we see substantial reductions in the counts of nouns, verbs, and adpositions. Notably, the reduction in adjective counts accounted for a larger proportion of Gemma's overall length reduction. This is because the original Gemma model would start the game with longer messages that contain more adjectives than humans. Because our post-training was not intended to alter the model behavior when initially mentioning something, the post-trained model’s messages still often contain adjectives in Repetition 1 (average adjectives count: 1.51). Humans, however, start with relatively concise language with an average of 0.48 adjectives in Repetition 1. Also, human messages more often contained pronouns in early repetitions, which are dropped later (pronoun count in Repetition 1:  0.71 (human), 0.46 (Gemma)). For Llama, nouns accounted for a greater proportion of shortening (34%), also because the post-trained Llama (just like the original Llama) started the interaction with more nouns than humans (average noun count 2.65 vs. 1.47).
> >
> > **Table 2**: POS counts reduction from Repetition 1 to Repetition 6 (proportion of total word reduction). Negative values mean the count increased.
> >
> > |  | NOUN | VERB | ADJ | ADP | PRON |
> > | :---- | ----- | ----- | ----- | ----- | ----- |
> > | Human | 15% | 13% | 7% | 14% | 18% |
> > | Gemma | 18% | 14% | 33% | 8% | 0% |
> > | Llama | 34% | 8% | 24% | 16% | \-3% |
> >
> > These analyses show that our post-trained models share several similarities with humans in how they develop more efficient language, though these models’ behaviors are also affected by the backbone model we started post-training with. Future research can focus on letting LLMs behave like humans even at the beginning of these conversational scenarios, a direction orthogonal to the convention formation behavior we focus on in this paper.
> >
> > \[1\]: Characterizing the Dynamics of Learning in Repeated Reference Games [https://onlinelibrary.wiley.com/doi/10.1111/cogs.12845](https://onlinelibrary.wiley.com/doi/10.1111/cogs.12845)

---

### Official Review · Reviewer_He5h · 2025-05-12

**Rating:** 6
**Confidence:** 3
**Ethics Flag:** 1

**Summary:**

This paper proposes a post-training method for efficient multi-turn conversation.
Focusing specifically on efficient reference convention formation, the authors first construct a dataset for coreference mention preference, and then formulate a new training strategy with newly introduced “mention planning tokens” and regularized SFT for the tokens.
For the evaluation, the authors used two tasks, text-only reference game, and document-grounded utterance completion, to investigate the effectiveness of the proposed method.
Human interaction data was also collected to quantify the communication efficiency of the current LLMs in the convention formation task.

**Questions To Authors:**

The dataset construction in section 3 seems to be based on some intuitive observations. It was not clear whether or not there is any bias in the dataset that makes the dataset different from the real human conversation. The claim of the paper is that the effectiveness of the model can be shown extrinsically by its effectiveness on other out-of-domain tasks. However, if the reference convention patterns can be manually analyzed and categorized, we may be able to gain deeper insights into what actually  happens in the human and model conversations.

The experiments exclude the cases of partial matches (e.g. dustpan --> pan), but in reality, this could happen frequently in daily conversation. What is the motivation for excluding these cases?

**Reasons To Accept:**

The convention formation task is novel and will help to draw the community's attention to the issues of measuring and enhancing the LLMs’ naturalness in multi-turn conversations.

**Reasons To Reject:**

While the reviewer appreciates the challenge and importance of the task,  the preferable approach was not necessarily clear for me.Table 2 shows that shortening the reference convention can make the conversation more efficient but can also reduce accuracy and success rates. Detailed explanation/analysis would be helpful

---

> ### Author Response · Authors · 2025-05-31
> **Author Response**
>
> Thank you for your comments and questions!
>
> **"The preferable approach was not necessarily clear for me… Table 2 shows that shortening the reference convention... Detailed explanation/analysis would be helpful"**
>
> An ideal model should show improved efficiency (shortening), improved consistency (reducing Word Novelty Distance (WND) across repetitions), and similar listener accuracy levels and trends as humans (improved accuracy across repetitions). These are accomplished with our method.
>
> Table 2 is our ablation study results. It is expected that some ablated versions of our method achieve slightly better performance on one of the metrics because they have similar ideas and training procedures as our main method. Critically, the ablated variant that achieved better listener accuracy than our main method is using excessively long messages toward the end, and **the accuracy gap is marginal and not statistically significant**. The conclusion from Table 2 is that none of the ablated variants can show consistent improvements across all the metrics and both evaluation tasks, and they each have substantial issues, whereas our full method achieves consistent improvements.
>
> A similar note regarding Figure 2 (our main results):
> Figure 2 shows that our post-trained models achieve significant improvement in efficiency and consistency. While they show small drops in listener accuracy compared with the corresponding off-the-shelf models, these drops are not statistically significant and the listener accuracy of the post-trained models improves quickly over time. Therefore, our approach takes a meaningful step towards solving this problem and will encourage future work in this direction (both based on our methods, and relying on our two accessible benchmarks).
>
> **"The dataset construction in section 3 seems to be based on some intuitive observations. It was not clear whether or not there is any bias in the dataset that makes the dataset different from the real human conversation."**
>
> You raise an important question regarding dataset bias. This would have been particularly concerning if our evaluation would have used this data. We were aware of this risk, and therefore designed our evaluation studies to be completely separate domains and scenarios. Specifically, in the case of the reference game, the evaluation is based on decades of research in cognitive science showing this scenario as an effective testbed for this behavior. Intuitively, it doesn’t matter if the training data has issues, as long as the evaluation is separate and focused on the targeted skill (and our targeting is based on CogSci research).
>
> Additionally, we avoid overfitting to the training data with careful regularization, during both SFT and preference optimization.
>
> The rationale behind our dataset construction was that multi-turn interactions (i.e., in TV scripts) are more likely to include convention formation in a way that is easy to detect and build a training data around. It’s also based on what data is publicly available in sufficient quantity to create a training dataset.
>
> **"If the reference convention patterns can be manually analyzed and categorized, we may be able to gain deeper insights into what actually happens in the human and model conversations."**
>
> This is an excellent suggestion. We will manually analyze and categorize the data and add a qualitative analysis section. If the paper is accepted, we get an extra page for the camera ready, so we can add it there.
>
> **"The experiments exclude the cases of partial matches (e.g. dustpan \--\> pan), but in reality, this could happen frequently in daily conversation. What is the motivation for excluding these cases?"**
>
> We prohibit these cases to make sure the reference task is non-trivial. This is a scenario design decision, because of our use of text and not images. We were concerned that if we allow these, a speaker may “cheat” by always using a lexeme of a compound word (i.e. pan) to refer to a target word (i.e., dustpan), thereby not really creating descriptive language, and stifling any chance for convention formation.

---

> > ### Comment · Reviewer_He5h · 2025-06-07
> > **Thank you for your response**
> >
> > Thank you for your detailed explanation. It has clarified the interpretation of Table 2.
> >
> > Your response has given me a clearer understanding of the challenges involved in collecting useful data for this task.
> >
> > The paper is inspiring in that it highlights the ability of LLMs in conversation formation. As this is pioneering work, I think there are limitations as well. For example, communication efficiency is limited to the length of referenced phrases. Additionally, it is impossible to determine from the performance values alone whether the dialogue is natural and accurate.
> > I believe the contribution could be further strengthened by additional analysis and discussion, particularly through the inclusion of negative examples.

---

> > > ### Author Response · Authors · 2025-06-09
> > > **Follow-up responses**
> > >
> > > Thank you for considering our work inspiring and pioneering! If you find our responses helpful, we will be very grateful if you reflect this in updating your rating.
> > >
> > > Below are our responses to the limitations you mentioned.
> > >
> > > **“Treating efficiency as length reduction”**
> > > Length reduction is not the only form of communication efficiency we examined. We also measured  language consistency. Consistency reflects efficiency as it reduces cognitive costs in both production (i.e., generation) and comprehension. This key factor of efficiency was also studied in prior work (Hua and Artzi, 2024), where it was shown as challenging for models. We agree that there are other ways to improve efficiency beyond convention formation. Our work enables and encourages future research in this area, by setting strong improvements over the current model behavior and providing the necessary evaluation framework (and data).
> > >
> > > **Additional analysis of language behaviors.**
> > >
> > > We observe that models can generate natural and accurate languages easily. Regarding **negative examples**, the most conspicuous ones are where our models do not show any shortening and just repeat the initial messages. Those messages are still natural and reasonable. These instances of repetition are also found with the original models and humans. A more subtle type of negative example is when the model message does describe the target but is not specific enough, given that we used similar items to form challenging referential contexts. We will provide more negative examples in our camera-ready version.
> > >
> > > As for **language patterns for convention formation**, we analyzed humans’ and Gemma’s changes from one repetition to another for 30 reference games (Table 1). The most common strategy for both humans and the post-trained Gemma is to remove a phrase from the previous message while keeping the rest of the message (“Drop Phrase”). The removed phrase is often a prepositional phrase or a clause. Another strategy is rephrasing, using a concise expression to replace part or all of a message while conveying a similar meaning. For example, the referring expression for a “waterbed” can change from “the mattress that is filled with liquid” to “the liquid-filled mattress”. Finally, both humans and Gemma sometimes replace the previous description by describing a new feature that is more concise. For example, a hammer can be initially described as “the tool that is used to fix things around the house” and later as “the tool that is used to build things”. These analyses show that humans and Gemma share similarities in their strategies, though Gemma’s more frequent use of new features likely has contributed to the slightly lower language consistency than humans (as our WND shows). We will add our analysis for Llama in our camera-ready draft as well.
> > >
> > > **Table 1**: Common Convention Formation Strategies
> > >
> > > |  | Drop Phrase | Rephrase | New Feature |
> > > | :---- | ----- | ----- | ----- |
> > > | Human | 66% | 19% | 15% |
> > > | Gemma (ours) | 62% | 11% | 26% |
> > >
> > > Additionally, we counted the POS tags for models and humans, looking at how the counts change over time (Table 2). For all systems, we see substantial reductions in the counts of nouns, verbs, and adpositions. Notably, the reduction in adjective counts accounted for a larger proportion of Gemma's overall length reduction. This is because the original Gemma model would start the game with longer messages that contain more adjectives than humans. Because our post-training was not intended to alter the model behavior when initially mentioning something, the post-trained model’s messages still often contain adjectives in Repetition 1 (average adjectives count: 1.51). Humans, however, start with relatively concise language with an average of 0.48 adjectives in Repetition 1. Also, human messages more often contained pronouns in early repetitions, which are dropped later (pronoun count in Repetition 1:  0.71 (human), 0.46 (Gemma)). For Llama, nouns accounted for a greater proportion of shortening (34%), also because the post-trained Llama (just like the original Llama) started the interaction with more nouns than humans (average noun count 2.65 vs. 1.47).
> > >
> > > **Table 2**: POS counts reduction from Repetition 1 to Repetition 6 (proportion of total word reduction). Negative values mean the count increased.
> > >
> > > |  | NOUN | VERB | ADJ | ADP | PRON |
> > > | :---- | ----- | ----- | ----- | ----- | ----- |
> > > | Human | 15% | 13% | 7% | 14% | 18% |
> > > | Gemma | 18% | 14% | 33% | 8% | 0% |
> > > | Llama | 34% | 8% | 24% | 16% | \-3% |
> > >
> > > These analyses show that our post-trained models share several similarities with humans in how they develop more efficient language, though these models’ behaviors are also affected by the backbone model we started post-training with. Future research can focus on letting LLMs behave like humans even at the beginning of these conversational scenarios, a direction orthogonal to the convention formation behavior we focus on in this paper.

---

> > > > ### Comment · Reviewer_He5h · 2025-06-10
> > > > **Thank you for your response.**
> > > >
> > > > Thank you for the discussion and further analysis!

---

### Official Review · Reviewer_uQr9 · 2025-05-13

**Rating:** 9
**Confidence:** 4
**Ethics Flag:** 1

**Summary:**

The paper investigates the ability of off-the-shelf LLMs to participate in human linguistic convention formation, more specifically the gradual shortening of referring expressions over the course of a dialog. A fine-tuning method using human data is proposed, implemented, and shown to lead to significant improvements.

**Questions To Authors:**

- Line 153, is (x, y1, yw) in the wrong order? (x, yw, y1) is used elsewhere,

- Line 214: "The second task is based on the Doc2Dial dataset (Feng et al., 2020), where a user is querying an agent in a document-grounded interaction." - is this referring to a human agent or an LLM (or some other type of articificial agent)? If the latter, how come it exhibits convention formation capabilities?

- Line 235: The win-rate judging model should be confirmed to agree with human judgements. Suggest to add this as future work.

- Line 244 onwards: In the data collection process, human users interact with an LLM, which (as shown in the paper) does not handle concept formation. Could this affect the data collected in some way? If not, why not?

- Line 252 onwards: How do you know for sure that all deviations from the expected reduction in WND was due to participants misunderstanding the instructions? Also, it is not entirely clear what the misunderstanding consists in. As it is, one may wonder if there may be some variation in human convention formation capabilities that is being disregarded here.

Finally, could it be worth trying zero shot prompting as a method to enable convention formation?

**Reasons To Accept:**

The topic is very relevant and has not been sufficiently studied. The paper is very clear and the work described is well carried out.

**Reasons To Reject:**

None.

---

> ### Author Response · Authors · 2025-05-31
> **Author Response**
>
> Thank you for your comments and questions\!
>
> **Line 153:**
> Thank you for spotting this typo. It’s the wrong order. It should be (x, yw, yl).
>
> **Line 214 (Doc2dial data):**
> For the data collection of the Doc2dial dataset (not collected by us), human crowd-sourcing workers played the user and agent roles, so convention formation behaviors are prevalent.
>
> **Line 235 (win-rate judging model's agreement with human):**
> We discuss win-rate agreement with human judgements in the appendix (lines 489-499). We will move it to Section 4 with the extra page allowed for camera-ready (if accepted). The competence rating based on the LLM judge (a binary score) achieves a Cohen’s kappa of 0.749, which is greater than the 0.6 threshold and indicates substantial inter-annotation agreement. The more fine-grained win/lose/tie labels achieve a Krippendorff alpha of 0.705, which is also greater than the 0.667 threshold to be acceptable for drawing tentative conclusions.
>
> **Line 244 onwards (potential impact of using an LLM listener):** While LLMs do not spontaneously show convention formation as a speaker, prior work ([Hua and Artzi, 2024](https://arxiv.org/pdf/2408.01417)\) shows that strong LLMs do show human-like trends when being a listener (improved accuracy over time). Therefore, the prior work uses a GPT4o listener to replace real human listeners. We also adopt this approach to reduce experiment costs. We did conduct a small-scale human-human pilot study where both the speaker and the listener are humans (Appendix A, Line 456-463). The human-human study's results are highly consistent with our human-model studies. We will further explain this design choice in our revised draft.
>
> **Line 252 onwards (How do you know for sure that all deviations from the expected reduction in WND was due to participants misunderstanding the instructions?):**
> This is an excellent point and has impact beyond LLM research. Indeed, there may be other modes/incentives of human behaviors that led to high WNDs. We speculate that it’s caused by misunderstanding of game instruction because many people expressed, in the exit survey, that they thought they were asked to come up with new descriptions every time, which our instruction does not. Because the human study in this paper is to establish a reasonable goal to align the model with, we took the behavior of the majority group of humans (the convention forming group).
>
> We think our findings of the high WND human group can encourage future research in cognitive science, which can use a larger pool of human participants to study nuanced reasons behind the less frequent phenomenon of inconsistent referring expressions. Even if this phenomenon is mainly caused by misunderstandings, it would be interesting to study the deep reasons behind the misunderstandings.
>
> It’s important to note that we were very careful not to introduce any form of priming in our instructions, not even mentioning that the study is about adaptation (something that has been done in past studies). This was important to observe natural behavior, but also led to ambiguity in a minority of cases.
>
> **“Could it be worth trying zero shot prompting”**
> This is a good point. We tried this approach and found that simply telling the models to shorten their messages over time will lead to poor message consistency in our text reference games. We find that only very explicit prompts on how to shorten the messages and maintain consistency can achieve human-like performance in text reference games. These prompts are very scenario-specific, and do not generalize beyond reference games. These observations are consistent with the findings on image reference games reported by [Hua and Artzi, 2024](https://arxiv.org/pdf/2408.01417)\. As this prior work also points out, the prompts needed to elicit human-like behaviors are too specific to generalize outside of the evaluation scenario, so we needed post-training as a more robust approach.

---

> > ### Comment · Reviewer_uQr9 · 2025-06-09
> >
> > Thank you for the clarifying responses.

---

### Official Review · Reviewer_XAPw · 2025-05-18

**Rating:** 6
**Confidence:** 4
**Ethics Flag:** 1

**Summary:**

This paper studies how to make large language models speak more like humans by learning to simplify their words over time in a conversation. The authors create a special training method that teaches models to reuse shorter phrases when talking about the same thing again. They also design two new tests to check if the model learned this behavior, and show big improvements. The work is original, well-structured, and addresses a problem not many others focus on. It is clear, thoughtful, and could inspire better communication in AI systems. However, the model does not truly understand why it should form conventions, and the training method is an add-on rather than built into the model's core training.

**Questions To Authors:**

1. In Figure 2 (right), the post-trained models do not reach the best accuracy compared to GPT-4o and Claude, even though they improve over time. Can you explain why your method leads to lower early accuracy?

2. Can your model decide when to use a short reference and when to give a full description, depending on the context? For example, if the short form is unclear or the user needs more detail, can the model switch back to using a full expression? This ability seems important in real-world use, but it is not discussed or evaluated in the paper.

**Reasons To Accept:**

- The paper gives a new and useful way to help language models talk more like humans by forming shorter and clearer phrases.
- It shows strong results on two tasks and may help future research on better model communication.

**Reasons To Reject:**

The proposed method helps the model speak more simply, but the model does not really understand why it should do this. It could just copy the patterns from training data without real reasoning. The authors do not discuss if the model can choose between using short or full expressions based on the situation. For example, in some cases, the user needs a full and clear answer to avoid confusion. The paper also does not check if the model can switch back to using full expressions when needed, such as when the short form is ambiguous. This is important for real-world use, but it is not evaluated in the paper.

---

> ### Author Response · Authors · 2025-05-31
> **Author Response**
>
> Thank you for your comments and questions\!
>
> **“The proposed method helps the model speak more simply, but the model does not really understand why it should do this.**”
> This is an important point that we address in part in Section 7\. We will further elaborate this discussion in the next version of the paper. Our method improves the convention formation behavior by letting the LLM better model the human behaviors in training data, which is a common approach for LLM training and post-training. We agree that this approach does not explicitly encourage “effort-reduction” considerations (one could hypothesize that these are acquired implicitly, but we are not in a position to make this claim). We believe instilling such *explicit* consideration is an important direction for future work. Critically, our work enables this, because we set strong improvements on vanilla model behavior and create the necessary evaluation framework (and data). We invested a lot of effort in setting the framework for easy comparisons, hoping that our contribution enables more work on this problem.
>
> Regarding the comment **“the training method is an add-on rather than built into the model's core training”**:
> Our method can be easily interleaved into LLMs’ overall post-training regimes, which are often considered as core part of training. This is because contemporary post-training methods often rely on many reward signals with different tuning methods (as mentioned briefly in this [OpenAI blog post](https://openai.com/index/sycophancy-in-gpt-4o/)). So, our process can be interleaved easily into such a multi-signal process.
>
> **“The authors do not discuss if the model can choose between using short or full expressions based on the situation … Question 2: For example, if the short form is unclear or the user needs more detail, can the model switch back to using a full expression?”**
> We discuss the basic ability to behave differently depending on the context: a good model should use a sufficiently detailed expression when it initially introduces a concept (before conventions can form) and only show shortening later (as conventions form). Our repeated reference game results show that the post-trained models indeed have this ability.
>
> The example you mention is an excellent extension and covers an important advanced ability beyond convention formation: switching back to elaborate language upon the listener’s confusion or when more information is needed. This would require separate evaluation scenarios where confusion and ambiguity are common. We opted to avoid such scenarios because it would dramatically complicate convention formation evaluation – our main goal in this paper. We think this ability can be better positioned in the orthogonal task of adapting model behaviors upon negative user feedback, but we do see the connection to our target behavior. Because our paper focuses on convention formation and is already very dense in content, we leave this direction to future work.
>
> **“The post-trained models do not reach the best accuracy compared to GPT-4o and Claude. Can you explain why your method leads to lower early accuracy”**
> The open-weight models we started post-training with had lower accuracy than the proprietary models (GPT-4o and Claude) in the first place. If we compare each post-trained model with its corresponding off-the-shelf open-weight counterpart, the early accuracy gaps are smaller and not statistically significant. We will clarify this in the paper.

---

### Decision · Program_Chairs · 2025-07-06

**Decision:**

Accept

**Comment:**

This paper offers a post-training approach to make large language models improve their communication efficiency over the course of multi-turn dialogues. Specifically, the approach aims to get models to follow a human-like approach of slowly making the referring phrases for an object shorter over time (e.g., “The round, white thing that is bigger than the plates but smaller than the pot” to “the round thing”). The post-training approach uses realistic training data that shows examples of these patterns in practice. Then the work applies the method to two base models and demonstrate improvements on two tasks inspired by prior literature.

All reviewers highlighted notable strengths of this work including:

    Novelty and importance of the research area (uQr9, He5h, Usy9), and that it may inspire community attention and future work in this area (XAPw, He5h)
    The technical approach was original (Usy9), well-structured (Usy9), well-executed (uQr9), and ultimately yielded strong experimental results (XAPw, Usy9)
    Solid evaluation methodology (XAPw, Usy9)
    Clear writing (XAPw, uQr9)

Reviewers mostly rated the paper as marginally above the acceptance threshold due to some remaining concerns:

    Efficiency may not equal improved task accuracy or success (He5h, Usy9, XAPw). Shortening may in some cases reduce task success as short/simple references may not always be appropriate or clear enough (humans may flexibly switch between short and long based on the context, but it is unclear if the approach can handle this).
    May be difficult to generalize for real world use given the post-training “add on” approach as it may not be sustainable to train new models for new situations (XAPw, Usy9)
    Lack of detailed analysis to inform future work (Usy9, He5h)
    Lack of knowledge about model’s understanding vs. just pattern matching (XAPw)
    Missing stronger inference-time baseline (e.g., simple prompting such as requesting the model be as concise as possible) (Usy9)

While some of these issues (1, 2, 4) are a bit more fundamental to the work and difficult to fix, the authors in the discussion promised to include more detailed analysis in the revised version (3) and addressed the point about the baseline (5). I agree with the reviewers that the proposed changes will make this paper stronger, and I find the paper to be an interesting and well-executed investigation into communication efficiency that is likely to inspire future work. I recommend accepting the paper.